# Interface engineering breaks both stability and activity limits of RuO$_2$ for sustainable water oxidation

Kun Du[1,6], Lifu Zhang[2,6], Jieqiong Shan[3], Jiaxin Guo[1], Jing Mao[1], Chueh-Cheng Yang[4,5], Chia-Hsin Wang [4] ✉, Zhenpeng Hu [2] ✉ & Tao Ling [1] ✉

Designing catalytic materials with enhanced stability and activity is crucial for sustainable electrochemical energy technologies. RuO$_2$ is the most active material for oxygen evolution reaction (OER) in electrolysers aiming at producing 'green' hydrogen, however it encounters critical electrochemical oxidation and dissolution issues during reaction. It remains a grand challenge to achieve stable and active RuO$_2$ electrocatalyst as the current strategies usually enhance one of the two properties at the expense of the other. Here, we report breaking the stability and activity limits of RuO$_2$ in neutral and alkaline environments by constructing a RuO$_2$/CoO$_x$ interface. We demonstrate that RuO$_2$ can be greatly stabilized on the CoO$_x$ substrate to exceed the Pourbaix stability limit of bulk RuO$_2$. This is realized by the preferential oxidation of CoO$_x$ during OER and the electron gain of RuO$_2$ through the interface. Besides, a highly active Ru/Co dual-atom site can be generated around the RuO$_2$/CoO$_x$ interface to synergistically adsorb the oxygen intermediates, leading to a favourable reaction path. The as-designed RuO$_2$/CoO$_x$ catalyst provides an avenue to achieve stable and active materials for sustainable electrochemical energy technologies.

The practical application of water electrolyser in the generation of sustainable green hydrogen energy[1–3] calls for the development of stable and active electrocatalysts. So far, RuO$_2$ is the most active electrocatalyst for anodic oxygen evolution reaction (OER) in water electrolysis[4–9]. Unfortunately, as indicated by Pourbaix diagram[10–12], RuO$_2$ is thermodynamically unstable under OER conditions over the entire pH range. This has been verified by extensive theoretical and experimental investigations[4,13–15], which demonstrate that the proceeding of OER is accompanied by the transformation of stable Ru$^{4+}$ to unstable Ru$^{n>4+}$, resulting in the gradual dissolution and deactivation of the catalyst. Common strategies of improving the stability of RuO$_2$ include mixing RuO$_2$ with a more corrosion resistant material in the synthetic procedure[6,16–19] and controlling the dispersion of RuO$_2$ to avoid direct contact with the electrolyte[20]. In these cases, however, the stability of Ru-based catalysts is generally enhanced at the expense of its activity, leading to a seesaw relation between stability and activity[14,21–26]. It is necessary to develop new strategy to achieve both enhanced stability and activity for Ru-based catalysts.

To substantially enhance the stability of RuO$_2$ catalysts under OER conditions, we identify that the key is to suppress the electrochemical corrosion of Ru species. There is a classic fashion of using a sacrifice component to protect the target material. For example, in the well-known zinc-plated steel[27], the more reactive zinc is preferentially oxidized to form a dense oxide film over the steel, preventing the further

[1]Key Laboratory for Advanced Ceramics and Machining Technology of Ministry of Education, Institute of New-Energy, School of Materials Science and Engineering, Tianjin University, Tianjin 300072, China. [2]School of Physics, Nankai University, Tianjin 300071, China. [3]School of Chemical Engineering and Advanced Materials, The University of Adelaide, Adelaide, SA 5005, Australia. [4]National Synchrotron Radiation Research Center, Hsinchu 30076, Taiwan, ROC. [5]Department of Materials Science and Engineering, National Yang Ming Chiao Tung University, Hsinchu 30010, Taiwan, ROC. [6]These authors contributed equally: Kun Du, Lifu Zhang. ✉e-mail: wang.ch@nsrrc.org.tw; zphu@nankai.edu.cn; lingt04@tju.edu.cn

oxidation of zinc and the corrosion of steel. Inspired by this, we assumed that implementing a proper material with $RuO_2$ to form a stable interface can be a promising strategy to stabilize $RuO_2$ catalyst. On the other hand, previous works of Nørskov et al.[14,15] have suggested that the 'stable' $RuO_2$ exhibits unsatisfactory catalytic activity due to the lack of unstable high-valence $Ru^{n>4+}$ species. Regarding this, the construction of an interface may create new active sites[28] to break the activity limit of 'stable' $RuO_2$. Moreover, the interface construction may use some cost-effective materials to reduce the use of precious metal Ru and achieve sustainable water electrolysis.

Herein, we report constructing a $RuO_2/CoO_x$ hybrid catalyst to break the stability-activity seesaw relation on $RuO_2$ catalyst. Combining theoretical calculations, in situ X-ray photoelectron spectroscopy (XPS) with in situ UV-visible (UV–Vis) absorption spectroscopy, we demonstrate that the stability of the new $RuO_2/CoO_x$ hybrid significantly exceeds the Pourbaix limits of bulk $RuO_2$. This is ascribed to the sacrificing oxidation of $CoO_x$ and interfacial electronic effects, which stabilized $RuO_2$ by decreasing driving force for $RuO_2$ dissolution and enriching electrons on $RuO_2$. In addition, as verified by kinetic isotope effect (KIE), in situ infrared reflection (IR) measurements and theoretical calculations, the construction of interface creates highly active Ru/Co dual-atom sites around the $RuO_2/CoO_x$ interface, which synergistically absorb the key oxygen intermediates during OER to optimize the reaction thermodynamics and kinetics. Therefore, the $RuO_2/CoO_x$ catalyst achieves superior high OER activities under neutral and alkaline conditions accompanied by excellent long-term stability.

## Results

### Stabilization of $RuO_2$ on $CoO_x$ support

According to our calculated Pourbaix diagram of $RuO_2$ (Fig. 1a), $RuO_2$ undergoes oxidation in the OER potential range, forming high-valence

$Ru^{n>4+}$ ions that dissolve in the electrolyte[4,13,29]. We assume that depositing $RuO_2$ on an appropriate support that can be preferentially oxidized represents a rational strategy to protect $RuO_2$ from dissolution in harsh electrochemical oxidation. To test this hypothesis, $CoO_x$ was selected as the support material, which is easily oxidized under the anodic potential in the OER range (Supplementary Fig. 1). The calculated Pourbaix diagram of $RuO_2/CoO_x$ (Supplementary Note 1) in near-neutral and alkaline environments is shown in Fig. 1b. As expected, the $CoO_x$ support is gradually oxidized from CoO to $Co_3O_4$, CoOOH and eventually $CoO_2$ with the increase of anodic potential. Hereafter, $CoO_x$ repents these cobalt oxides for simplicity. Significantly, $RuO_2$ can construct stable interfaces with the oxidation products of $CoO_x$ (CoO, $Co_3O_4$, CoOOH, and $CoO_2$) within the entire OER potential range (Fig. 1b, c). Besides, stable Ru–O–Co chemical bond can be formed at the $RuO_2/CoO_x$ interface (Fig. 1c and Supplementary Fig. 2), which enables the hybrid to gain considerable energy from constructing the interface (Supplementary Fig. 2). This undoubtedly lowers the energy of the hybrid system and decreases the driving force for $RuO_2$ dissolution, thus stabilizing $RuO_2$ in the hybrid catalyst.

To further understand the interfacial effect on stabilizing $RuO_2$, Bader charge analysis was performed on four $RuO_2/CoO_x$ catalysts, i.e., $RuO_2/CoO$, $RuO_2/Co_3O_4$, $RuO_2/CoOOH$, and $RuO_2/CoO_2$. As shown in Fig. 1d–f, the changes in the charges of Ru, O and Co ions at the interface relative to those in their corresponding bulk materials show a similar trend among the four catalysts. Taking $RuO_2/CoOOH$ as an example, the average charge of Ru ions away from the interface in $RuO_2$ is ~6.3 $e$, which increases to 6.7 $e$ at the interface (Fig. 1d), indicating the enrichment of electrons on the interfacial Ru ions. Similarly, the average charge of O ions in the bulk $RuO_2$ is ~6.6 $e$, which increases to ~6.7 $e$ at the interface, and further increases to ~7.0 $e$ in the bulk CoOOH (Fig. 1e). Note that the Co charge at the

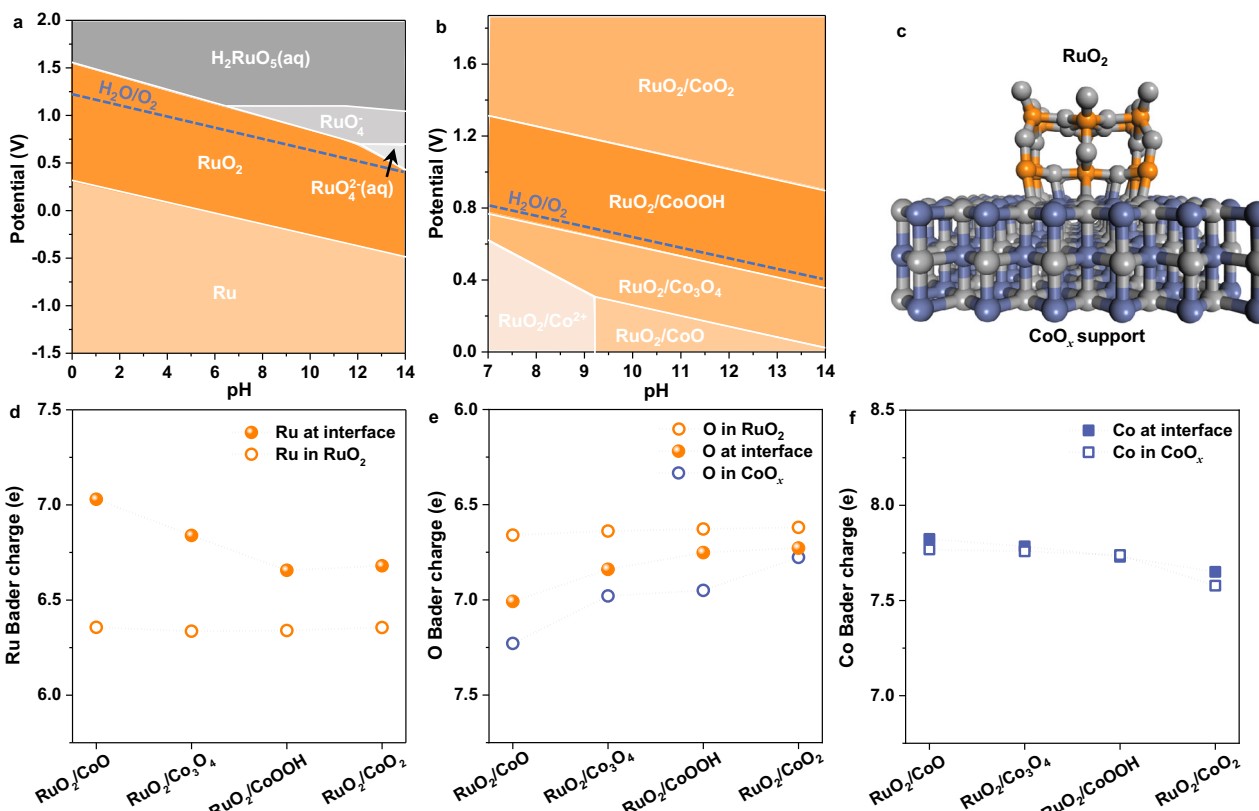

**Fig. 1 | Investigation on stability of RuO2 on CoOx support. a, b** Calculated Pourbaix diagrams of $RuO_2$ and $RuO_2/CoO_x$, respectively. Ion concentrations are $10^{-6}$ M. The potentials in **a** and **b** are referenced to standard hydrogen electrode (SHE). **c** Schematic diagram of the interfacial structure of $RuO_2/CoO_x$. **d–f** Bader charges of the interfacial Ru, O and Co ions and their counterparts in $RuO_2/CoO$, $RuO_2/Co_3O_4$, $RuO_2/CoOOH$ and $RuO_2/CoO_2$, respectively.

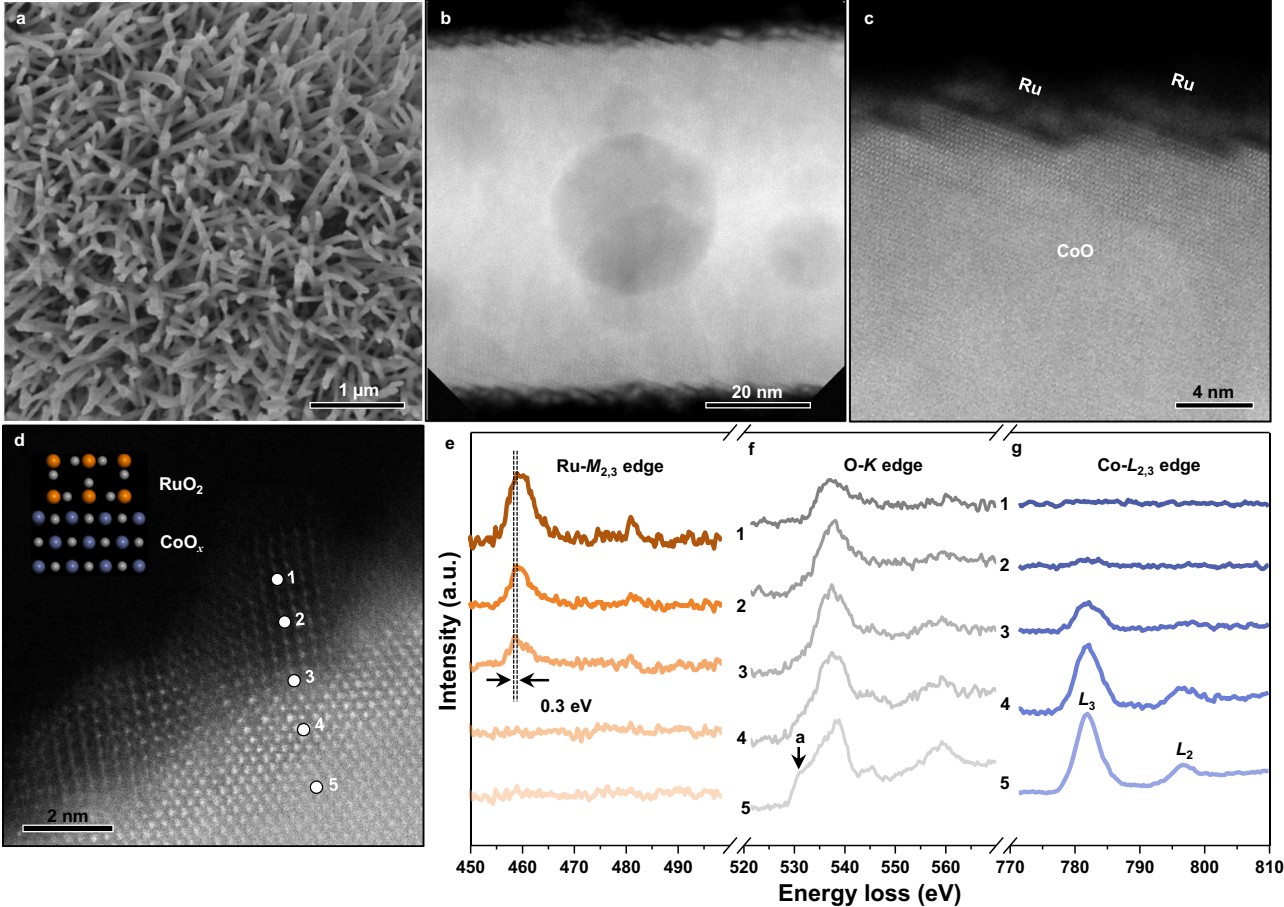

**Fig. 2 | Synthesis of RuO2/CoOx hybrid catalyst. a** Low-magnification SEM image of Ru/CoO. **b**, **c** Low- and high-magnification HADDF-STEM images of Ru/CoO, respectively. **d**, Atomic resolution HADDF-STEM image of RuO$_2$/CoO$_x$, with the inset showing the atomic model of RuO$_2$/CoO$_x$. **e–g** EELS spectra of Ru-$M_{2,3}$, O-$K$, and Co-$L_{2,3}$ edge across the interface from point 1 to point 5 in **d**, respectively.

interface is almost identical to that in the bulk CoOOH (Fig. 1f). These results indicate that O ions in the hybrids play a key role in the electron enrichment in interfacial Ru ions. This is due to the different metal-oxygen hybridizations in RuO$_2$ and CoOOH, resulting in different O charges in these two materials. That is, the O ions connecting with Co ions own more electrons compared with those connecting with Ru ions. Once Ru−O−Co bond is formed at the RuO$_2$/CoO$_x$ interface, the electron-rich O ions connecting with Co ions contribute electrons to the nearby Ru ions through metal-oxygen re-hybridization, thus enriching electrons in the interfacial Ru ions.

## Synthesis of RuO$_2$/CoO$_x$ hybrid catalyst

Guided by the above theoretical findings, RuO$_2$/CoO$_x$ hybrid catalyst was fabricated by depositing Ru nanoparticles on CoO nanorods (Fig. 2a), followed by an electrochemical oxidization process (Supplementary Figs. 3–8). As shown in Fig. 2b, c, the CoO nanorods possess faceted surface with prefabricated nanoscale roughness to uniformly load Ru nanoparticles. The Ru nanoparticles form a fish scale-like single-layer with a thickness of 2 nm on the surface of CoO nanorods (Supplementary Fig. 5). Subsequent electrochemical oxidation resulted in in situ conversion of Ru to RuO$_2$ on CoO$_x$ nanorods (Supplementary Figs. 7 and 8). This method features the epitaxial growth of RuO$_2$ on CoO$_x$ nanorods (Fig. 2d), providing a structural basis for strong interfacial geometric and electronic interaction between RuO$_2$ and CoO$_x$. The as-formed interface was closely inspected by sub-ångstrom resolution aberration corrected high-angle annular dark-field scanning transmission electron microscopy

(HAADF-STEM, Fig. 2d and Supplementary Fig. 9), showing an atomic-level tight connection of Ru, O and Co atoms at the interface. This finding was supported by the Fourier transform extended X-ray absorption fine structure (FT-EXAFS) of RuO$_2$/CoO$_x$ (Supplementary Fig. 10).

Electron energy-loss spectroscopy (EELS) at Ru-$M_{2,3}$, O-$K$, and Co-$L_{2,3}$ absorption edges was performed to investigate charge changes of Ru, O and Co ions across the interface (from point 1 to point 5) in Fig. 2d. As illustrated in Fig. 2e, the collected Ru-$M_{2,3}$ spectrum at the interface (point 3) shifts 0.3 eV toward the low energy loss direction with respective to that of RuO$_2$ (point 1), indicating a decreased Ru valence at the interface. For O-$K$ edge spectra (Fig. 2f), the curves show obvious shape change from RuO$_2$-like (point 1) to CoO$_x$-like (point 5). In particular, the characteristic peak 'a' collected in CoO$_x$ gradually weakens towards the interface until disappears in RuO$_2$. This reflects different electronic properties of O atoms connecting with Ru and Co atoms, respectively, and re-hybridization of O atoms at the interface caused by simultaneous connection with Ru and Co atoms. Notably, no noticeable peak shift is observed in the collected Co-$L_{2,3}$ spectra (Fig. 2g and Supplementary Fig. 11). These experimental results well support the calculated evident charge change of O ions from CoO$_x$ to RuO$_2$ via the interface (Fig. 1e), while no significant Co charge change from bulk CoO$_x$ to the interface (Fig. 1f). This indicates that O ions play a decisive role in the reduction of Ru valence through the electronic interaction among Ru, O and Co atoms at the interface. We note that the enrichment of Ru charge at the interface will affect the distribution of Ru charge in the bulk and on the surface through continuous Ru−O bonds.

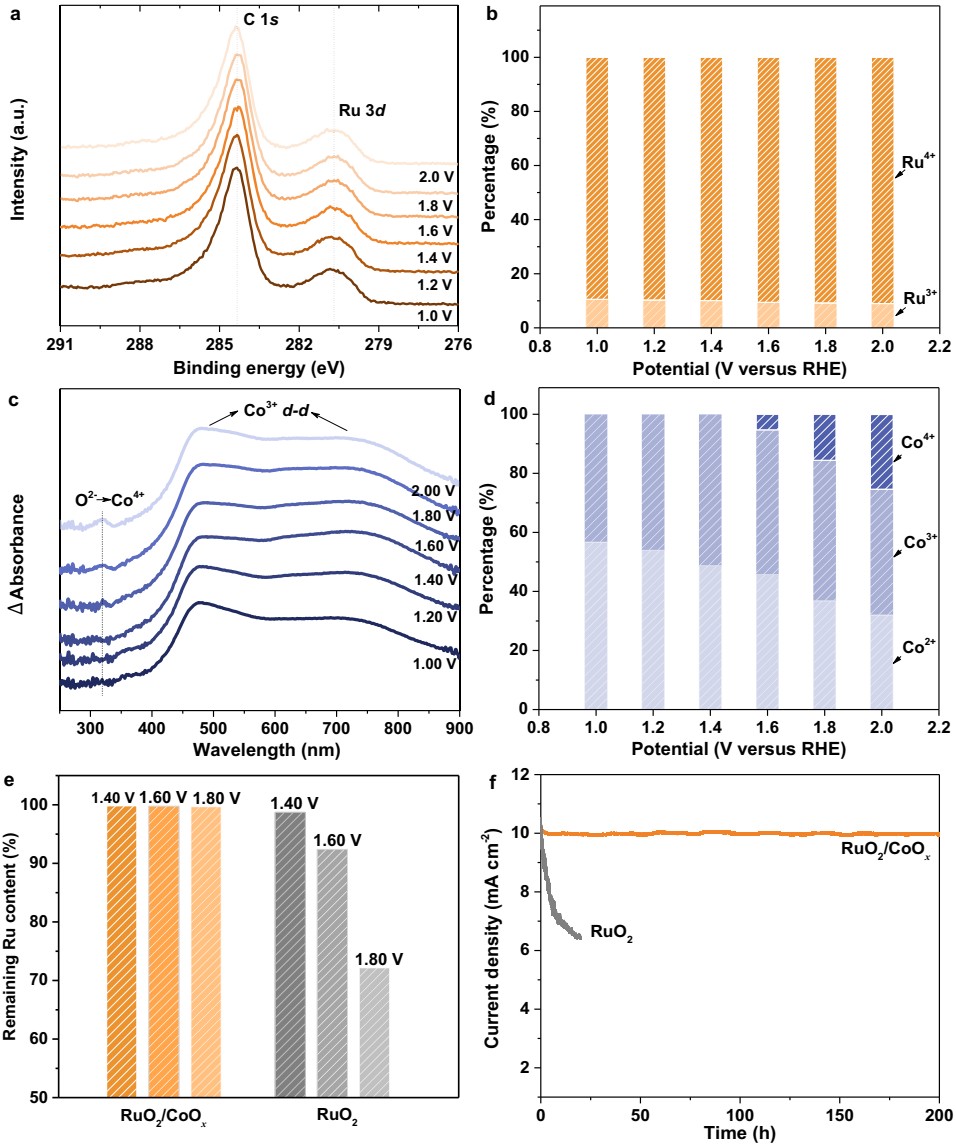

**Fig. 3 | Investigation on in situ stability of RuO2/CoOx hybrid catalyst during OER. a**, **b** In situ Ru 3$d$ XPS spectra recorded at applied potential during 1.00–2.00 $V_{RHE}$ and corresponding $Ru^{3+}$ and $Ru^{4+}$ content ratios, respectively. **c** In situ UV–Vis spectra recorded at applied potential during 1.00–2.00 $V_{RHE}$. The spectrum at each potential was collected by subtracting the spectrum of pristine RuO2/CoOx. **d** $Co^{2+}$,

$Co^{3+}$ and $Co^{4+}$ content ratios in RuO2/CoOx under different applied potentials based on EPR analysis. **e** Retention ratios of Ru in RuO2/CoOx and RuO2 after continuously tested at different potentials for 20 h. **f** Potentiostatic tests of RuO2/CoOx (at 1.47 $V_{RHE}$) and RuO2 (at 1.59 $V_{RHE}$) for an initial current density of 10 mA cm$^{-2}$.

## Stability evaluation of RuO2/CoOx in OER

Afterwards, the stability of RuO2/CoOx hybrid catalyst during OER in neutral environment was monitored by in situ XPS (Supplementary Figs. 12 and 13). Significantly, the Ru 3$d$ XPS peak at 280.9 eV exhibits negligible changes with the applied potential increased from 1.0 to 2.0 V versus reversible hydrogen electrode (RHE) (Fig. 3a). Detailed quantitative analysis shows the co-existence of $Ru^{3+}$ and $Ru^{4+}$ species with almost identical percentages from 1.0 to 2.0 $V_{RHE}$ (Fig. 3b and Supplementary Fig. 14). Surprisingly, even at 2.0 $V_{RHE}$, there is still 9% of $Ru^{3+}$ remaining in the RuO2/CoOx hybrid. Considering that the average particle size of RuO2 is ~2 nm, the theoretical proportion of interfacial Ru atoms to total Ru atoms should be about 15% (Supplementary Note 2 and Supplementary Fig. 15). This value is in agreement with the percentage of $Ru^{3+}$ species as demonstrated by the in situ XPS results (Fig. 3b), indicating the critical role of the constructed interface in stabilizing RuO2 in the hybrid.

Moreover, although the Ru valence state of RuO2/CoOx hybrid did not exceed 4+ in the studied potential range, the Co valence state

increased significantly during OER as evidenced by in situ UV–Vis spectroscopy characterization and quantitative electron paramagnetic resonance (EPR) analysis. It was demonstrated that as the anodic potential increased, the Co ions in the hybrid catalyst underwent gradual oxidation from $Co^{2+}$ to $Co^{3+}$ and $Co^{4+}$ without dissolution (Fig. 3c, d, Supplementary Fig. 16 and Supplementary Table 1). This is consistent with the calculated Pourbaix diagram of the hybrid catalyst (Fig. 1b) and verified our hypothesis that the support CoOx was preferentially oxidized to protect RuO2.

The above in situ spectroscopic results were supported by the experimentally observed remarkable stability of RuO2/CoOx during OER. As shown in Fig. 3e, Supplementary Figs. 17 and 18 and Supplementary Table 1, after 20 h continuous stability test at the potential as high as 1.80 $V_{RHE}$, the content of Ru element in the hybrid catalyst was still close to 100%. Significantly, the RuO2/CoOx catalyst works stably at a constant current density of 10 mA cm$^{-2}$ for more than 200 h (Fig. 3f), and affords an excellent dynamic stability with varied current density from 10 to 100 mA cm$^{-2}$ (Supplementary Fig. 19). In sharp contrast, the

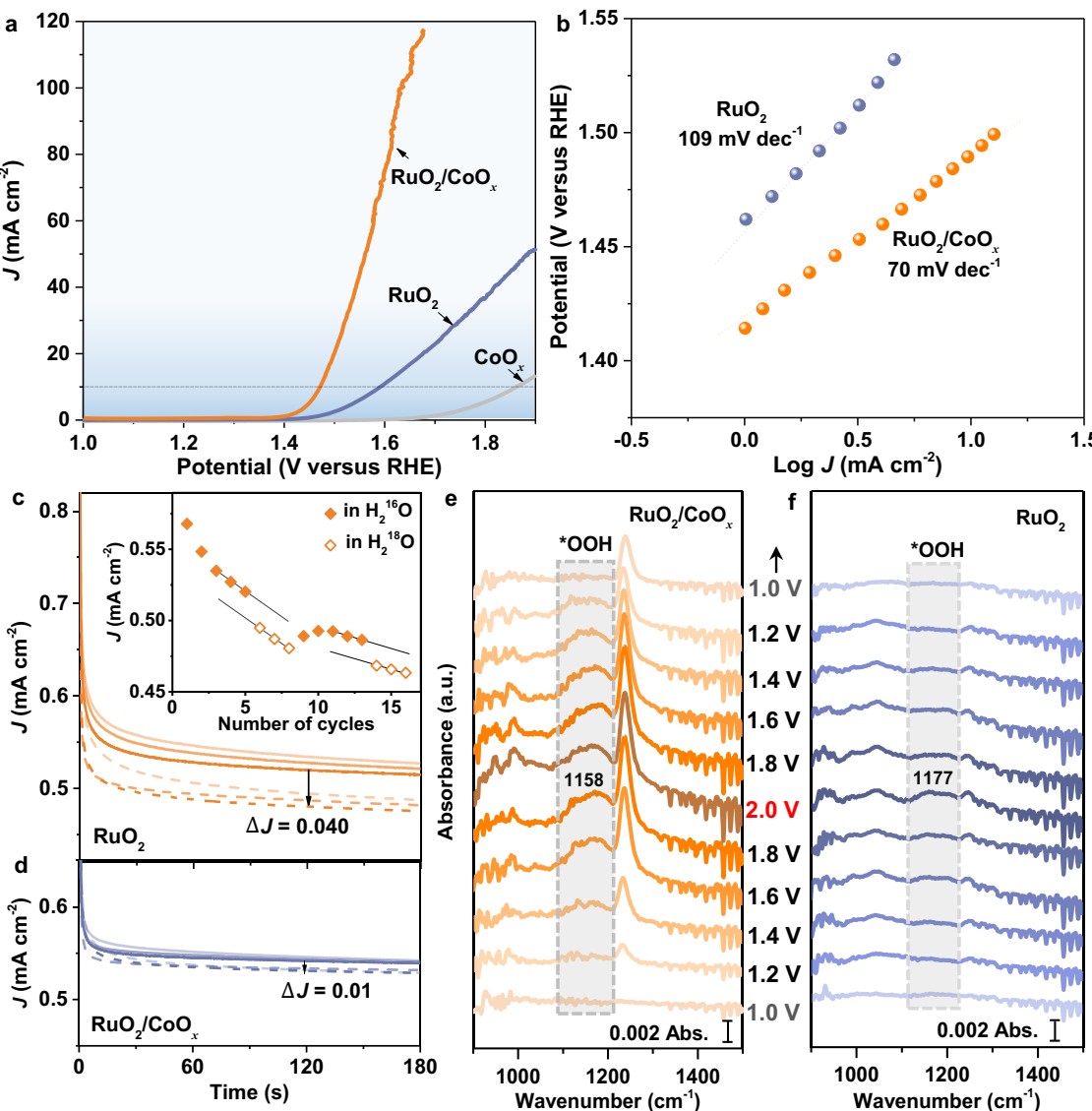

**Fig. 4 | OER performance and RDS of RuO2/CoO$x$ hybrid catalyst. a, b** OER polarization curves of RuO$_2$/CoO$_x$, RuO$_2$ (deposited on carbon black) and CoO$_x$, and corresponding Tafel curves, respectively. **c, d** Current density ($J$)-time curves of RuO$_2$ and RuO$_2$/CoO$_x$ with multiple cycles in either H$_2^{16}$O (solid lines, closed triangles) or H$_2^{18}$O (dashed lines, open triangles), respectively. The inset in **c** represents the average current density ($J_{average}$) during the last 2 minutes for each cycle. **e, f** In situ surface-enhanced IR spectra of RuO$_2$/CoO$_x$ and RuO$_2$ at different potentials. The potentials were referenced to RHE. Note that the catalysts here were tested in neutral electrolyte and the RuO$_2$ loading on CoO$_x$ is 10 μg on per cm$^2$ electrode.

pristine RuO$_2$ (deposited on carbon black, Supplementary Figs. 20 and 21) encountered severe catalyst dissolution and performance degradation (Fig. 3f and Supplementary Fig. 22), which agrees well with the literature[4,13,29]. Additionally, the RuO$_2$/CoO$_x$ also demonstrated excellent stability in alkaline environment (Supplementary Figs. 23, 24 and Supplementary Table 2).

## OER activity and rate-determining step of RuO$_2$/CoO$_x$

Under the incentive of the high stability, we evaluated the OER activity of the RuO$_2$/CoO$_x$ hybrid catalyst with a RuO$_2$ mass loading of 10 μg on per cm$^2$ electrode (Supplementary Table 3, Supplementary Figs. 25 and 26). Note that RuO$_2$ (Supplementary Fig. 27 and Supplementary Table 4) and CoO$_x$ catalysts were measured as control samples. As shown in Fig. 4a and Supplementary Fig. 28, the RuO$_2$/CoO$_x$ exhibits a much higher OER activity than RuO$_2$ and CoO$_x$ in neutral electrolyte, affording an ultra-low overpotential of 0.24 V to drive an OER current density of 10 mA cm$^{-2}$. Besides, the current density of RuO$_2$/CoO$_x$ can achieve 400 mA cm$^{-2}$ at 1.92 V$_{RHE}$ when the mass of RuO$_2$/CoO$_x$ catalyst

is increased to 1.5 mg cm$^{-2}$ on nickel foam (Supplementary Fig. 29). Impressively, the RuO$_2$/CoO$_x$ is amongst the most active OER catalysts reported so far under neutral conditions (Supplementary Table 5). Moreover, the turnover frequency (TOF) of the RuO$_2$/CoO$_x$ was estimated by normalizing the O$_2$ generation rate to the total number of Ru ions on CoO$_x$ support (Supplementary Note 3). At an overpotential of 400 mV, the RuO$_2$/CoO$_x$ delivers a high TOF of 3.61 s$^{-1}$, representing a 10-time enhancement in comparison with the optimum value reported previously on Ru-based catalyst (RuIrCaO$_x$[30], 0.36 s$^{-1}$). Moreover, the RuO$_2$/CoO$_x$ achieves a high OER Faradaic efficiency of ~98% at 10 mA cm$^{-2}$ (Supplementary Fig. 30).

To reveal the activity origin of the RuO$_2$/CoO$_x$, we explored the rate-determining step (RDS) of OER by Tafel plots. As illustrated in Fig. 4b, the RuO$_2$/CoO$_x$ shows a significantly decreased Tafel slope (70 mV dec$^{-1}$) compared with the RuO$_2$ (109 mV dec$^{-1}$), indicating the possible different RDSs in these two catalysts. $^{18}$O/$^{16}$O isotope effect[31] was then employed in both catalysts to probe the O–O bond formation, which is generally considered as the RDS in OER[32,33]. As shown in

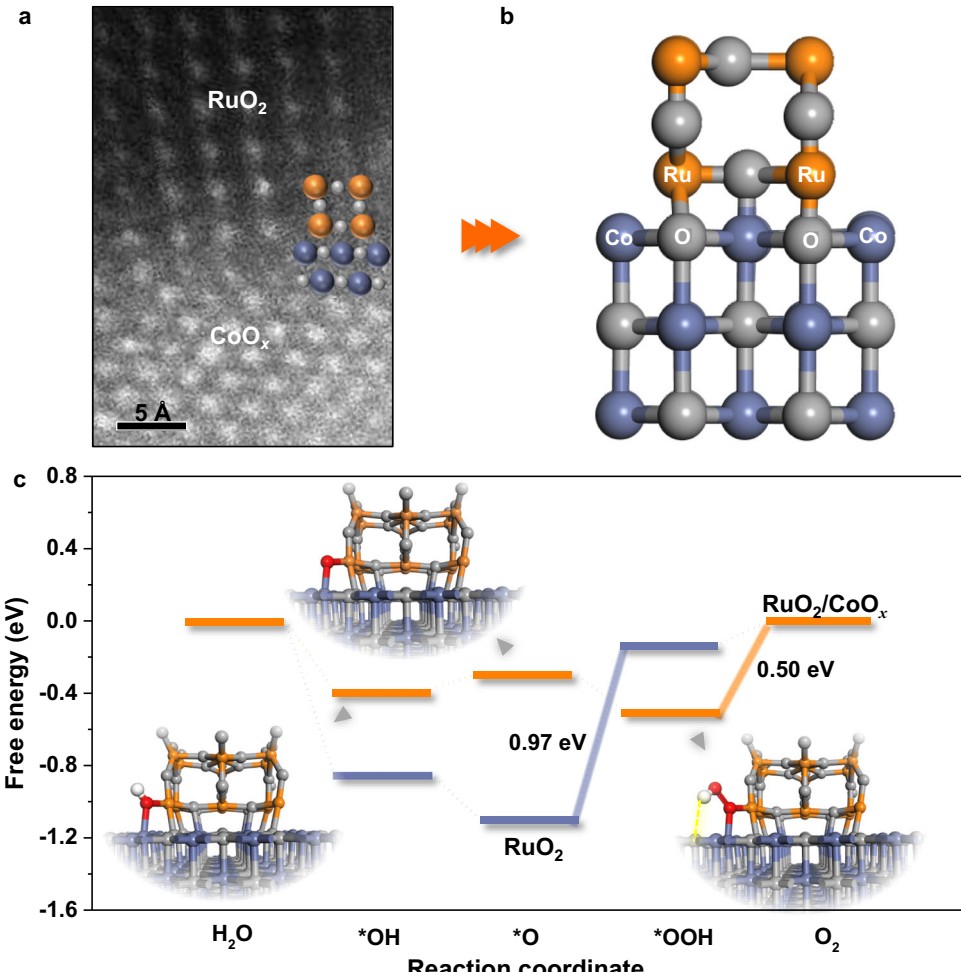

**Fig. 5 | Theoretical study on origin of enhanced activity on RuO2/CoO$x$.**
**a** HADDF-STEM characterization of RuO$_2$/CoO$_x$ interface exposed active sites.
**b** Schematic diagram of the structure model of RuO$_2$/CoO$_x$ hybrid catalyst.
**c** Calculated OER free energy diagrams for RuO$_2$ and RuO$_2$/CoO$_x$, with the inset
showing the computationally-optimized geometric structures of *OH, *O and *OOH
intermediates co-adsorbed on Ru/Co dual-atom site exposed around the RuO$_2$/
CoO$_x$ interface.

Fig. 4c, there is an obvious decrease in the OER current density on the RuO$_2$ catalyst when the electrolyte was changed from H$_2^{16}$O to H$_2^{18}$O, and the KIE value of the O−O bond formation step (KIE$_{O-O}$) is estimated as 1.03 (Supplementary Fig. 31). Since the KIE$_{O-O}$ value falls within the range between 1.01 and 1.04[34,35], the O−O bond formation step can be confirmed to be the RDS of the RuO$_2$. In contrast, the negligible Δ*J* between H$_2^{16}$O and H$_2^{18}$O for the RuO$_2$/CoO$_x$ demonstrates that O−O bond formation is not the RDS (Fig. 4d). This finding is further supported by the in situ IR spectroscopy characterization (Fig. 4e, Supplementary Fig. 32 and Supplementary Table 6), which shows a more pronounced *OOH band of RuO$_2$/CoO$_x$ in comparison with that of RuO$_2$ (Fig. 4f). These results suggest that the RuO$_2$/CoO$_x$ exhibits a different RDS compared with the pristine RuO$_2$ as we will discuss in detail later.

Furthermore, we demonstrate that the RuO$_2$/CoO$_x$ hybrid catalyst delivers a superior high OER performance in alkaline environment, permitting it a promising candidate for highly efficient OER electrocatalysts in a wide pH range (Supplementary Fig. 33 and Supplementary Table 7).

### Origin of enhanced OER activity on RuO$_2$/CoO$_x$

A key question remains how the RuO$_2$/CoO$_x$ interface significantly boosts the OER activity of RuO$_2$. To shed light on this, density functional theory (DFT) calculations were performed. In particular, HADDF-STEM imaging (Fig. 5a) shows that Ru/Co dual-atom sites were exposed

around the RuO$_2$/CoO$_x$ interface after treating the hybrid at the OER onset potential (-1.40 V$_{RHE}$). Accordingly, the computational model was constructed (Fig. 5b). It was found that the exposed Ru/Co dual-atom site around the interface is the most active site for OER (Supplementary Figs. 34–36); the oxygen intermediates, i.e., *OH, *O and *OOH, tend to be co-adsorbed at the Ru/Co dual-atom site to form a stable nearly quadrilateral structure (Fig. 5c, inset).

Significantly, the triatomic *OOH bents downward and the H atom forms a hydrogen bond with the surface O in the CoO$_x$ to construct a unique *OO−H···O adsorption configuration. Due to the electrostatic attraction of O atom in the CoO$_x$, the O−H bond length in the formed *OOH increases compared with that on the pristine RuO$_2$ (Supplementary Fig. 35). According to previous work[36], when the intermolecular hydrogen bond stretches the bond in the probe molecule, it will lead to a shift of the stretching vibrational frequency of the probe groups toward the low wavenumber direction in IR spectra. Relative shift of *OOH bands is observed in the in situ IR spectra of the RuO$_2$/CoO$_x$ compared with those of RuO$_2$ (Fig. 4e, f), verifying the adsorption configuration of *OO−H···O, which facilitates the stabilization of *OOH at the Ru/Co dual-atom site around the interface (inset of Fig. 5c and Supplementary Fig. 35).

Note that *OOH is a key intermediate during OER, which exhibits a high formation barrier and restricts the OER activity of catalysts[32,33]. The calculated Gibbs free energy for *OOH formation (Δ$G_{*OOH}$) on the RuO$_2$

is as high as 1.12 eV (Fig. 5c and Supplementary Fig. 35). Notably, this calculated value of $\Delta G_{*OOH}$ is consistent with the result reported by Nørskov et al. and other researchers[15,24], indicating an inferior OER activity of 'stable' $RuO_2$ with the absence of the generated high-valent $Ru^{n>4+}$ species during OER[4,13,29]. As expected, for the $RuO_2/CoO_x$, the *OOH formation is greatly facilitated at the Ru/Co dual-atom site around the interface. More importantly, this shifts the RDS of $RuO_2/CoO_x$ to the subsequent step of *OOH formation – that is, desorption of *$O_2$ (Supplementary Note 4), which demonstrates a significantly decreased energy injection of 0.50 eV (Fig. 5c). This exciting finding agrees well with the KIE and in situ IR results (Fig. 4c–f). Therefore, our well-consistent experiments and calculations confirm that the artificially constructed $RuO_2/CoO_x$ hybrid catalyst successfully breaks the OER activity limit of 'stable' $RuO_2$ by changing the RDS of OER through exposing the highly active Ru/Co dual-atom sites around the $RuO_2/CoO_x$ interface (Supplementary Note 5 and Supplementary Figs. 37–39).

## Discussion

In summary, we constructed the $RuO_2/CoO_x$ hybrid catalyst to break the stability and activity limits of $RuO_2$ by decoupling its stability-activity relation. Specifically, the sacrificial oxidization of $CoO_x$ and the electron interaction among the face-to-face Ru–O–Co interfacial atoms enhance the stability, while the Ru/Co dual-atom site exposed around the interface is responsible for the improved activity. With such unique electronic and geometric effects generated by the $RuO_2/CoO_x$ interface, we solved the critical issues of $RuO_2$ under OER conditions and achieved high stability and excellent activity. Our work provides an atomic scale understanding of employing interfacial effect to simultaneously enhance the stability and activity of $RuO_2$. We believe that under the guideline built by the $RuO_2/CoO_x$ interface, the activity and stability issues of $RuO_2$ in acidic environments can also be fundamentally solved by selecting appropriate support materials. We expect that this work will also contribute to future research on other renewable energy technologies coupled with OER in neutral environments, such as reduction of carbon dioxide to multi-carbon fuels.

## Methods

### Synthesis of $RuO_2/CoO_x$ and $RuO_2$ catalysts

$RuO_2/CoO_x$ catalyst was synthesized by in situ electrochemical transformation method with Ru/CoO as the starting material. Briefly, CoO nanorod arrays were first fabricated on carbon fiber paper or fluorine-doped tin oxide (FTO) substrates by cation exchange methodology[37,38]. Afterwards, ruthenium precursor solution was prepared by dissolve $RuCl_3$ in ethanol/water ($V_{ethanol}/V_{water} = 1:1$) to achieve a 30 mM $RuCl_3$ solution. Then, CoO nanorods were immersed in 40 mL of ultrapure water, and an appropriate amount of ruthenium precursor solution was added, aged for 6 h, dried at room temperature, and finally heated by $N_2$ flow at 400, 500 and 550 °C for 0.5 h to obtain $RuO_2$ with average particle sizes of 2, 3 and 4 nm, respectively (Supplementary Fig. 38). Note that the Ru loading mass on CoO nanorods can be easily controlled by tuning the adding volumes of ruthenium precursor solution. Finally, the obtained Ru/CoO nanorods were electrochemically oxidized by scanning cyclic voltammetry between 0.80-1.50 $V_{RHE}$ to attain $RuO_2/CoO_x$ catalysts (Supplementary Fig. 8). The loading mass of $RuO_2$ on $CoO_x$ after optimization is 10 µg on per $cm^2$ electrode (Supplementary Fig. 26 and Supplementary Table 3). For the synthesis of $RuO_2$ reference catalyst, a similar method was applied using carbon black as the support material. The loading mass of $RuO_2$ on carbon black after optimization is 84 µg on per $cm^2$ electrode (Supplementary Fig. 27 and Supplementary Table 4). $RuO_2$ with this loading mass was characterized in Figs. 3 and 4 as reference sample.

### Materials characterization

Scanning electron microscopic (SEM) and transmission electron microscopic (TEM) images were performed on a Hitachi S-4800

SEM and a JEOL 2100 TEM, respectively. HAADF-STEM images were collected on a JEOL ARM200F microscope with a STEM aberration corrector operated at 200 kV. The convergent semi angle and collection angle were 21.5 and 200 mrad, respectively. EELS spectra were collected using a Titan Themis Cubed G2 60-300 operated at 200 kV. EPR measurements were carried out on a JEOL JES-FA200. The inductively coupled plasma mass spectrometry (ICP-MS) measurements were performed on an Agilent 7700x. X-ray diffraction (XRD) characterization was carried out on a Bruker D8 Advance diffractometer with Cu Kα radiation. The X-ray absorption fine structure spectra of Ru $K$-edge were performed at 4B9A beamline in Beijing Synchrotron Radiation Facility (BSRF). The storage rings of BSRF was operated at 2.5 GeV with a stable current of 400 mA. The OER Faradaic efficiency of $RuO_2/CoO_x$ was measured by a gas chromatograph (GC-2014, Shimadzu, Japan) equipped with a thermal conductivity cell detector.

### In situ spectroscopic characterizations

In situ XPS spectra were measured by ambient pressure XPS end station equipped with a static electrochemical cell at NSRRC TLS BL24A (Supplementary Fig. 12a). The counter electrode was a Pt wire and the reference electrode was a Pt wire coated with Ag/AgCl paste. The working electrode was a carbon paper loaded with $RuO_2/CoO_x$ catalysts, which was cut into a circle with a diameter of 5.5 mm. During in situ XPS test, both the counter and reference electrodes were immersed in the electrolyte and sealed by a Nafion membrane and the carbon paper was sandwiched between the Nafion membrane and a Ta foil for electrical contact[39,40] (Supplementary Fig. 12b). The analysis chamber pressure is around 0.3 mbar due to water diffusing onto the sample's surface and evaporating into the chamber while in situ XPS spectra were measured.

In situ UV–vis spectroscopy was performed on a Hitachi U-3010 with a homemade photo-electrochemical cell, with catalysts fabricated in situ on a FTO substrate as the working electrode, a Pt wire as counter electrode and an Ag/AgCl electrode as the reference electrode.

In situ attenuated total reflectance surface-enhanced IR spectra were collected on a Fourier transform infrared spectrometer (Nicolet IS50, Thermo Fisher Scientific Co., Ltd) with a MCT detector and a Pike Technologies VeeMAX III ATR accessory. A catalyst ink was prepared by mixing 2 mg of catalyst investigated with 1 mL of ultrapure water and then deposited on an Au film coated Si prism. The Si prism, a Pt foil and an Ag/AgCl electrode were served as the working electrode, the counter electrode, and the reference electrode, respectively, in an H-type electrochemical cell, which was separated by a Nafion 115 membrane. All background curves were collected without applied potential in $N_2$-saturated electrolyte, and all spectra were collected with a 4 $cm^{-1}$ resolution.

### EPR tests

$RuO_2/CoO_x$ was treated at 1.00, 1.20, 1.40, 1.60, 1.80 and 2.00 $V_{RHE}$ for 5 min, respectively, and then dried quickly by high-purity $N_2$ (99.999%). Then, the treated catalysts were collected and transferred to an EPR tube under $N_2$ atmosphere. Then, the tube was immediately frozen and stored at 77 K using liquid nitrogen. The EPR measurement was performed at a modulation amplitude of 0.8 mT, a modulation frequency of 100 kHz, a conversion time of 50 ms and a time constant of 50 ms. During test, the temperature was set at 70 K. Quantitative analysis was conducted by double integration after baseline correction[41].

### KIE measurements

According to previous literature[31,34,35,42], multicycles chronoamperometric tests were carried out in 1.0 M phosphate buffered saline (PBS) with $H_2^{16}O$ and $H_2^{18}O$. The KIE value was estimated from the following

equation:

$$KIE = J_{H_2^{16}O} / J_{H_2^{18}O} \qquad (1)$$

where $J_{H_2^{16}O}$ and $J_{H_2^{18}O}$ are the average current density in $H_2^{16}O$ and $H_2^{18}O$, respectively. The average current density values of multicycles were linear fitted. The KIE value was estimated from the ratio of the two data points in the two fitted line in $H_2^{16}O$ and $H_2^{18}O$ (Supplementary Fig. 31).

## Electrochemical characterizations

The electrochemical performance of the catalysts in neutral (1.0 M PBS) and alkaline (1.0 M KOH) electrolytes was tested in a three-electrode system. A catalyst ink was prepared by ultrasonically dispersing 2 mg of catalyst, 2 mg of conductive carbon (Vulcan XC 72), 20 μL of 5 wt% Nafion solution and 20 μL of isopropanol in ultrapure water to achieve a catalyst concentration of 5 mg mL$^{-1}$. 10 μL of as-prepared catalyst ink was then dropped onto a polished glassy carbon rotating electrode (5 mm in diameter, Pine Research Instrumentation) serving as the working electrode (Supplementary Tables 3 and 4). The counter electrode was a Pt wire and the reference electrode was a calomel electrode saturated in KCl. The electrochemical tests were performed in $O_2$-saturated electrolyte with the working electrode rotating at a speed of 1600 rpm. All potentials were referenced to the RHE by using pure hydrogen calibration and corrected with 75% $IR$ loss, and all polarization curves were obtained with a scan rate of 5 mV s$^{-1}$.

## Computational methods

All spin-polarized DFT calculations were performed using Vienna *Ab initio* Simulation Package (VASP)[43-46]. The projector augmented wave (PAW) potentials[47] and Perdew-Burke-Ernzerhof (PBE) exchange-correlation functional[48] were adopted in the calculations with a plane wave kinetic energy cut-off of 400 eV. The energy converge criteria was set to be $10^{-4}$ eV, and the maximum force was converged to less than 0.05 eV Å$^{-1}$ on each ion. An effective $U$ parameter of 3.7 eV was applied for Co $3d$ states to describe well the electronic structure of CoO, $Co_3O_4$, CoOOH, and $CoO_2$[38]. For the computational model of $RuO_2/CoO_x$, the optimized lattice constants are $a = b = 18.10$ Å, $c = 26.40$ Å; for $RuO_2/Co_3O_4$, $a = b = 16.80$ Å, $c = 27.43$ Å; for $RuO_2/CoOOH$, $a = b = 17.30$ Å, $c = 32.07$ Å; for $RuO_2/CoO_2$, $a = b = 17.06$ Å, $c = 31.54$ Å. $K$-spaces were sampled using a $1 \times 1 \times 1$ grid. The free energy $(\Delta G)$ was computed from the following equation:

$$\Delta G = \Delta E + \Delta ZPE - T\Delta S - eU \qquad (2)$$

where $\Delta E$ is the energy difference of a given reaction, $\Delta ZPE$ is the zero-point energy correction, $\Delta S$ is the vibrational entropy change at a given temperature $T$, $e$ is the elementary charge, and $U$ is the electrode potential.

## Data availability

The data that support the findings of this study are available from the corresponding author on reasonable request.

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

## Acknowledgements

T.L. acknowledged funding from the National Natural Science Foundation of China (52071231and 51722103) and the Natural Science Foundation of Tianjin city (19JCJQJC61900). Z.P.H. acknowledged funding from the National Natural Science Foundation of China (21933006 and 21773124) and the Fundamental Research Funds for the Central Universities Nankai University (No. 63213042, 63221346, and ZB22000103). Calculations were performed on Supercomputing Center of Nankai University (NKSC) and TianHe-1A at the National Supercomputer Center, Tianjin.

## Author contributions

T.L. conceived the project, designed the experiments, and wrote the manuscript. K.D. and J.X.G. performed the experiments. L.F.Z. constructed models and conducted the DFT calculations guided by Z.H., and Z.H. designed some experiments to verify the correlation between theoretical models and experimental observations. C.Y. and C.W. performed the in situ XPS measurements. J.Q.S. commented and revised the manuscript. J.M. carried out the TEM and HADDF-STEM characterizations. All authors discussed the results and commented on the manuscript.

## Competing interests

The authors declare no competing interests.
