## [Peer review file · Nature Communications]

REVIEWER COMMENTS

Reviewer #1 (Remarks to the Author):

This paper proposes a resourceful method to improve the RuO₂ electrocatalyst activity and stability via the interfacial interaction with a supporting material, CoO_x. In general the approach is interesting and the combined experimental/theoretical work is good. However, there are a few comments that the authors should address before the manuscript can be considered for publication.

1) The authors demonstrate that the improvement is reached at the Ru/Co dual sites. However, they do not consider the relative amount of active sites w.r.t. to the Ru and Co sites in the pristine regions, as those Ru/Co dual sites appear to be highly dependent on synthesis conditions and on RuO₂ growth control. It would be good if the authors explain if the improvement gained on those local dual sites is enough to surpass the lower activity on the pristine ones.

2) A more specific comment is regarding the adhesion/formation energies discussed in lines 77 and 78 regarding SI Figure 2. Supplementary Figure 2 shows the theoretical investigation on the formation energy of the RuO₂/CoO_x hybrids, but not actual adhesion energy of RuO₂ complex on CoO_x supports. Each CoO_x support from (a) to (c) should have different formation energies, independently of their interaction with RuO₂. So, It is not clear if the trend represents stronger adhesion energy as proposed or just more energetically stable supports.

3) Regarding the model used for the computational analysis. Why is a nanoscale cluster used to represent RuO₂? Wouldn't be better to use a composite slab (top RuO₂, bottom CoO₂) to represent the interface between these materials? Please explain the rationale for the model used.

Reviewer #2 (Remarks to the Author):

The electrocatalytic splitting of water has drawn great interest these years. This paper reports the RuO₂/CoO_x hybrid catalyst to break the stability and activity limits of RuO₂ for stable and efficient OER catalyst in neutral and alkaline media. The sacrificial oxidization of CoO_x and the electron interaction among the face-to-face Ru–O–Co interfacial atoms enhance the stability, while the Ru/Co dual-atom site exposed around the interface is responsible for the improved activity. The authors have delivered detailed characterizations and thorough analyses of those catalysts, which provides an atomic-scale understanding of the interfacial effect of Ru–O–Co sites for simultaneously enhancing the stability and activity of RuO₂. Overall, the paper is well written and the experiments are carefully performed.

However, I think the total novelty of this work is not enough, as a similar approach has been published by Yu et al. (ACS Sustainable Chem. Eng. 2020, 8, 17520–17526) using an amorphous CoO_x decorated crystalline RuO₂ nanosheet catalyst, which is also a highly active and stable OER catalyst under alkaline condition. The author should explain the novelty of their approach as compared to Yu et al. We will reconsider this draft after addressing all comments.

1. Page 4 “Note that the change in the Co charge at the interface relative to the value in the bulk CoOOH is negligible (Fig. 1f).” Why? the difference of Co bader charge between bulk and interface can be negligible?
2. Authors mentioned that “Once Ru–O–Co bond is formed at the RuO₂/CoO_x interface, the electron-rich O ions connecting with Co ions contribute electrons to the nearby Ru ions through metal-oxygen re-hybridization, thus enriching electrons in the interfacial Ru sites” The Ru–O–Co bond only improves the stability of interfacial RuO₂, how this effect can be extended into the dominated RuO₂ surface (surface provides more active sites towards OER) since the limited interfacial bondings. Not only the interfacial species are active sites, but outer surface RuO₂ also. How can you distinguish those?
3. Authors roughly conclude that the EESL analysis supports the calculated charge changes, especially O ions from CoO_x to RuO₂ via the interface, please clearly describe.
4. From the in situ IR spectroscopy, the O–O bond formation can be confirmed as RDS on RuO₂, and a different OER mechanism occurs on RuO₂/CoO_x, what exact mechanism is? Please emphasize here.
5. Since the Ru–Co dual-atoms sites are served as co-works active sites in simulations analysis, the evidence of the corresponding experiments discovery cannot be missed.
6. Page 8 “Relative shift of *OOH bands is observed in the in situ IR spectra of the RuO₂/CoO_x compared with those of RuO₂ (Fig. 4e, f),” Please mark the indicated bands in the corresponding figures.
7. “..., verifying the adsorption configuration of *OO–H...O, which facilitates the stabilization of *OOH at the Ru/Co dual-atom site around the interface (inset of Fig. 5c and Supplementary Fig. 27).” As mentioned above, how to experimentally prove these configurations on Ru–Co dual-sites?
8. We believe that the Ru–Co dual sites practically work as synergistic catalysis, but the portion of interfacial atoms are limited what about else Ru species and substrate CoO_x? Do those non-interfacial species also catalyze OER? Some control experiments are missed.
9. The information about Ru–O–Co bonds should be additionally characterized and provided.

Reviewer #3 (Remarks to the Author):

Manuscript by Du et al. describes the synthesis of RuO₂ nanoparticles deposited on bulk CoOx support and application of this material for the alkaline oxygen evolution reaction. The study discusses an interesting concept and is generally carefully done, therefore can be recommended for publication in Nature Communications once the following issues are addressed.

- Intrinsic activity of the materials (normalized to the BET/electrochemically active surface area) rather than specific activity should be compared to prove that the electronic effects of the CoOx support rather than enhanced dispersion of the active species (compared to bulk RuO₂) are responsible for the activity improvement (RuO₂/CoOx vs. CoOx).

-Turnover frequency (P. 7) cannot be calculated based on Ru content only as there are no strong evidences that Ru is the only reaction site.

- Citation [30] does not refer to the study describing ¹⁶O/¹⁸O isotope effect (P. 7).

- I recommend to refrain from using phrases like “innovatively constructed”, “dream RuO₂”, “unprecedented high stability” to highlight the properties of the material.

- Fig. 1b: Ru species are missing from the low potential/pH region; Fig. 3f: what are the potentials applied?

Reviewer #4 (Remarks to the Author):

Dear authors

My review is in the document attached.

Best

Report on the paper NCOMMS-22-07796

General comments on the manuscript:

The manuscript describes the synthesis of a new RuO₂-based electrocatalyst consisting of RuO₂/CoO_x hybrid for the OER with high activities in both neutral and alkaline conditions. This paper is based on the suppression of the electrochemical corrosion of Ru species by fabricating a RuO₂/CoO_x hybrid to limit the dissolution of Ru species through the oxidation of Co-based species which are more stable. The paper presents a lot of sophisticated techniques combined with DFT to account for their claims but, in my opinion, the materials characterizations are not very convincing to account for the theoretical description of it. As a consequence, the theoretical results are very interesting but it seems they may not reflect the actual behavior of these materials. Moreover, as stated below, no mention of the acidic conditions are made where this strategy would really have been an asset to stabilize Ru-based catalysts so they could substitute Ir-based electrocatalysts. In general, many questions arise from the experimental part and how the authors draw conclusions from the experimental data.

My opinion is that the paper should be revised prior to publication and maybe in a more catalysis-oriented journal.

Some remarks about the experimental details

- 1) The use of CoO_x is somehow confusing: is it CoO with vacancies, does it represent the CoO precursor being oxidized into different others structure with different oxidation number for Co? For instance in the caption of figure 2: is it CoO or CoO_x?
- 2) How the value of the concentration for the Pourbaix diagram (10⁻⁶ M) was chosen and why is it similar for the calculation of Co-based, Ru-based and Co/Ru-based materials?
- 3) How is defined the formation energy of RuO₂/CoO_x hybrids (equation and free Gibbs energy definition)?
- 4) **Typo:** On the SI, supplementary note 2. Estimation **of** the proportion
- 5) The authors mentioned two types of substrates: Carbon fiber paper and FTO. Does the choice of the substrate has an influence on the growth of the CoO_x rods and the morphology of the samples? References 10.1038/ncomms12876 and 10.1038/s41467-017-01872-y only mention carbon fiber paper, stainless steel and carbon nanotube films and one can see differences in the density of the forest of rods and maybe length and diameter of the rods?
- 6) We can see some darker circular stains on the CoO_x nanorods (fig 2b, fig. S14a), from where does it come?
- 7) Fig S4 : what is the substrate (FTO or CFP)?
- 8) Fig S7 shows the electrochemical oxidation of Ru/CoO_x between 0.8 V and 1.5 V vs RHE while it is said to be between 1.0 and 1.5 V in the experimental part? Which one is the good value? It is the same range for RuO₂ only? It seems that in one case, the potential is set to a value where the catalytic regime occurs while in the other case no (cf fig. 4a), does it have an impact on the catalysis after wards?
- 9) It is possible to compare two systems which do not have the same morphology (RuO₂ cf. fig 21 and RuO₂/CoO_x, cf fig 2) but the authors should give some impacts of the morphology on the electrochemical activity and compare data for a same given mass or given surface area since the RuO₂ reference is not a commercial sample
- 10) Supplementary tables 3 and 4: I do understand how the RuO₂ wt% is calculated thanks to the footnote a but then the RuO₂ wt% in table 4 is calculated from the mass of the CFP support so why the calculation on table 3 does not include the support? Why do the authors use a wt% and not an atomic percentage?
- 11) On fig 3a, it seems that the peak for the C1s is shifted to lower energy while this of Ru 3d (3d_{5/2}) does not move. My question is: how were those spectra calibrated in energy since usually the C1s peak, related to adventitious carbon most of the time, serves as the reference?
- 12) No deconvolution of the XPS spectra are presented for the Ru³⁺ and Ru⁴⁺ ratio determination and even for all the element while a lot of literature exist on that subject. This is a missing aspect. A very good reference for Ru is [10.1002/sia.5852](https://doi.org/10.1002/sia.5852).

- 13) Neutral conditions were achieved by the use of PBS buffer, how the presence of phosphate species at the surface of the catalyst (in particular on RuO₂ fig 4f) reported through *in situ* surface enhanced IR spectra interfere with the catalysis process?

More general comments

- 14) The description of all the materials does not seem very clear and hence rather incomplete. In particular, with the data provided, it seems that the modelling of the RuO₂/CoOx hybrid is not in agreement with the experiments. On fig. S5, we can see a classical TEM image of Ru/CoO and EDS elemental mapping showing that Ru is dispersed throughout all the surface of the rod and so I cannot relate to the reported description of the system as islands of RuO₂ on a CoOx surface. In particular the HAADF-STEM (and not HADDF-STEM) presented in fig. S14a shows not a significant contrast that should arise from the difference in atomic number between Ru and Co. It is true they are both converted in oxides but still. So I think a HAADF-STEM images of the Ru/CoOx precursor is needed because I have the feeling that all the surface is covered by Ru particles and so the effect of the interface/triple point on the catalytic activity should be limited (cf paragraph "Origin of the enhanced OER activity"). The analysis of the fig 5a is deceiving in my point of view because the figures 2c and 2d are much in favor of a continuous Ru shell at the surface of the CoO rod. Could the authors provide more details on the actual description of their samples and hybrids, with more experimental evidences, since this a new material described?
- 15) Considering the fact that interfacing of RuO₂ with CoOx leads to Ru atoms with a higher electron density (cf fig 1d and 2e), this seems to be good for improving the stability but how is it better for improving the oxidation of water of hydroxide ions? I might miss something but Co charge does not change and the mechanism is said to occur at the interface (Ru/Co dual-atom site) between the two materials.
- 16) Is there any strains or defects induced by the transformation of Ru nanoparticles into RuO₂ nanoparticles at the interface between Ru and CoOx because we only see one detail of this interface (same fig S8, S9 and S12)?
- 17) RuO₂ is mainly a crucial catalyst for OER in acidic conditions since very efficient PGM-free catalysts are already available for alkaline OER. I regret that there is no mention of what happens in acidic condition (and the Pourbaix Diagrams in Fig. 1a and 1b are a bit deceiving in this sense because one goes to pH = 0 and the second only to 7: the calculations are not valid when pH<7?) Is this strategy of not applicable at low pH? The system seems anyway to be a bit complex to be included in a real electrolyzing cell and dissolution of the CoOx in low pH for instance may degrade the membrane.
- 18) Page 7, line 167-169, one can read "*This finding is further supported by the in situ IR spectroscopy characterization (Fig. 4e and Supplementary Table 6), which shows a more pronounced HOO* band of RuO₂/CoO_x in comparison with that of RuO₂ (Fig. 4f). These results suggest that the RuO₂/CoO_x exhibits a different OER mechanism compared with the pristine RuO₂.*" By checking in references 33, 38 and 39 (in the SI), it seems there is no value in the literature corresponding for this vibration in the 1150-1180 cm⁻¹ range which is indicated in the supporting table 6 and highlighted in the figures 4e and 4f. So the assignment of the OOH* band seems doubtful. First the fact that the OOH* intermediate is adsorbed on RuO₂/CoOx and not on Pt or Pd should provide a different wavelength number (it seems it is close from 1212 cm⁻¹ in reference 34 in the SI) so why the 1158 and 1177 cm⁻¹ are assigned to this specific vibration because I see no such values in the reference papers cited?

Isotopic measurements with H_2^{18}O could have been fruitful to gain evidence for this attribution. Does the phrase “the band is more pronounced” suggest that there is a quantitative difference (more or less adsorbates) and so a difference in the OER mechanism ?

Response to Reviewer #1

General Comments:

This paper proposes a resourceful method to improve the RuO₂ electrocatalyst activity and stability via the interfacial interaction with a supporting material, CoO_x. In general the approach is interesting and the combined experimental/theoretical work is good. However, there are a few comments that the authors should address before the manuscript can be considered for publication.

Response:

We would like to thank the Reviewer for his/her valuable comments and positive recommendation.

Original comment 1-1:

The authors demonstrate that the improvement is reached at the Ru/Co dual sites. However, they do not consider the relative amount of active sites w.r.t. to the Ru and Co sites in the pristine regions, as those Ru/Co dual sites appear to be highly dependent on synthesis conditions and on RuO₂ growth control. It would be good if the authors explain if the improvement gained on those local dual sites is enough to surpass the lower activity on the pristine ones.

Response:

Thanks for your valuable comment. The relative low activity of CoO_x compared with those of RuO₂/CoO_x and RuO₂ excludes pristine Co site as the most active OER site. We performed additional experiments to verify Ru/Co dual-atom sites as the most active OER sites. First, we deposited RuO₂ nanoparticles on carbon black with the same particle size (~2 nm) as that on RuO₂/CoO_x (Fig. R1a and R1b). The higher current density of RuO₂/CoO_x compared with that of RuO₂ with the same RuO₂-mass (Fig. R1c) indicates the crucial role of the interface in enhancing OER performance of RuO₂/CoO_x. Second, we prepared RuO₂/CoO_x with varied RuO₂ particle size or loading masses (Fig. R2a). Assuming either surface Ru site or interfacial Ru-Co dual-atom site as the most active sites, the corresponding site number was calculated and correlated with the measured OER current density at 1.50 V_{RHE} (Fig. R2b and R2c). An adequate linear relationship was observed between the number of interfacial Ru-Co dual-atom sites and the OER current density (Fig. R2c). These collective results reveals that the interfacial Ru/Co dual-atoms are the most active OER sites on RuO₂/CoO_x.

We would like to say that although the number of interfacial atoms is relatively small compared with that of surface Ru and Co sites on pristine RuO₂ and CoO_x, respectively, the OER mechanism on these interfacial sites is different from that on pristine RuO₂ and CoO_x. This is the origin of the significantly enhanced performance of RuO₂/CoO_x.

Figure R1. (a) TEM image of RuO₂ deposited on carbon black. (b) Polarization curves of the RuO₂ with different RuO₂ mass loadings on per cm² electrode in neutral electrolyte. (c) Comparison of current density (J) of RuO₂/CoO_x and RuO₂ at 1.50 V_{RHE} with identical RuO₂-masses.

Figure R2. (a) Polarization curves of the RuO₂/CoO_x with varied RuO₂ particle size or loading masses in neutral electrolyte. (b) and (c) Plot of current density (J) for RuO₂/CoO_x at 1.50 V_{RHE} as a function of the estimated number of surface Ru sites and interfacial Ru/Co dual-atom sites, respectively.

In response, we have added Figs. R1 and R2 as **Supplementary Figs. 34 and 36** in the revised supporting information. Moreover, **Supplementary Fig. 35** was added to show the characterization of RuO₂/CoO_x with different RuO₂ particle sizes. In addition, **Supplementary Note 4** was added to show details of Experiments that validate Ru/Co dual site as the active site for RuO₂/CoO_x.

“Supplementary Note 4. Experimental validation of Ru/Co dual site as the active site for RuO₂/CoO_x.”

First, we deposited RuO₂ nanoparticles on carbon black with the same particle size (~2 nm) as that on RuO₂/CoO_x (Supplementary Fig. 19c). As shown in Supplementary Fig. 34b, RuO₂/CoO_x exhibits a much higher current density than RuO₂ with the same RuO₂-mass loading, indicating the crucial role of interface in enhancing OER performance. Second, we prepared RuO₂/CoO_x with varied particle sizes (Supplementary Fig. 35) or loading masses of RuO₂. Assuming either surface Ru site

or interfacial Ru-Co dual-atom site as the most active sites, the corresponding site number can be calculated. Specifically, the number of interfacial Ru/Co dual-atom sites ($N_{\text{interfacial Ru/Co dual-atom site}}$) can be numerically calculated by

$$N_{\text{interfacial Ru/Co dual-atom site}} = N_{\text{RuO}_2} \times n_{\text{interfacial Ru/Co dual-atom site}} \quad (16)$$

where N_{RuO_2} is the total number of RuO_2 nanoparticles deposited on CoO_x , and $n_{\text{interfacial Ru/Co dual-atom site}}$ is the number of interfacial Ru/Co dual-atom sites on each RuO_2 particle.

Assuming that the RuO_2 nanoparticles are cube-shaped with an average side length (a) of 2 nm, N_{RuO_2} can be calculated by

$$N_{\text{RuO}_2} = \frac{m_{\text{RuO}_2}}{a^3 \times \rho} \quad (17)$$

where m_{RuO_2} is the loaded RuO_2 -mass and ρ is the density of RuO_2 . Moreover, $n_{\text{interfacial Ru/Co dual-atom site}}$ can be obtained by

$$n_{\text{interfacial Ru/Co dual-atom site}} = 4 \times (a/r_{\text{Ru-Ru}}) \quad (18)$$

where $r_{\text{Ru-Ru}}$ is the nearest neighboring Ru-Ru distance measured in supplementary Fig. 15. The number of surface Ru sites ($N_{\text{surface Ru site}}$) can be numerically calculated by

$$N_{\text{surface Ru site}} = N_{\text{RuO}_2} \times n_{\text{surface Ru site}} \quad (19)$$

where $n_{\text{surface Ru site}}$ is the number of surface of Ru sites on each RuO_2 particle, which can be calculated by

$$n_{\text{interfacial Ru/Co dual-atom site}} = 5 \times (a/r_{\text{Ru-Ru}})^2 \quad (20)$$

We correlated $N_{\text{interfacial Ru/Co dual-atom site}}$ and $N_{\text{surface Ru site}}$ with the OER current density of these samples at 1.50 V_{RHE} (Supplementary Fig. 36). An adequate linear relationship was observed between $N_{\text{Ru/Co dual-atom sites}}$ and the OER current density (Supplementary Fig. 36c). Therefore, these collective results reveals that the interfacial Ru/Co dual-atoms are the most active OER sites.”

Original comment 1-2:

A more specific comment is regarding the adhesion/formation energies discussed in lines 77 and 78 regarding SI Figure 2. Supplementary Figure 2 shows the theoretical investigation on the formation energy of the $\text{RuO}_2/\text{CoO}_x$ hybrids, but not actual adhesion energy of RuO_2 complex on CoO_x supports. Each CoO_x support from (a) to (c) should have different formation energies, independently of their

interaction with RuO₂. So, it is not clear if the trend represents stronger adhesion energy as proposed or just more energetically stable supports.

Response:

Sorry for misleading as we didn't give out the definition of the formation energy in the previous version. The "formation energy" was calculated by the following equation:

$$\Delta E = E_{\text{RuO}_2/\text{CoO}_x} - E_{\text{RuO}_2} - E_{\text{CoO}_x}$$

Here, the formation energy is the same as the adhesion energy defined in the literature (*Nat. Energy* 2017, 2, 17070).

According to the comment of the Reviewer, we have revised the related description in the **caption of Supplementary Fig. 2**.

"Supplementary Figure 2. Theoretical investigation on the adhesion energy of the RuO₂ cluster on CoO_x substrates. (a) RuO₂/Co₃O₄. (b) RuO₂/CoOOH. (c) RuO₂/CoO₂. Note that the adhesion energy (ΔE) was calculated by

$$\Delta E = E_{\text{RuO}_2/\text{CoO}_x} - E_{\text{RuO}_2} - E_{\text{CoO}_x} \quad (21)$$

where $E_{\text{RuO}_2/\text{CoO}_x}$, E_{RuO_2} , and E_{CoO_x} are the calculated energies for RuO₂/CoO_x, RuO₂ and CoO_x, respectively. The negative values of ΔE indicate a strong adhesion of RuO₂ to the CoO_x support. This undoubtedly decreases the driving force for RuO₂ dissolution, thus stabilizes RuO₂ in the hybrid catalyst."

Original comment 1-3:

Regarding the model used for the computational analysis. Why is a nanoscale cluster used to represent RuO₂? Wouldn't be better to use a composite slab (top RuO₂, bottom CoO₂) to represent the interface between these materials? Please explain the rationale for the model used.

Response:

Thanks for the comment. Here the main issue is the lattice mismatch between RuO₂ and Co based supports. For example, the lattice constants of CoO (100) surface model are $a = b = 3.017 \text{ \AA}$, and those for RuO₂ (110) are $a = 3.123 \text{ \AA}$, $b = 6.395 \text{ \AA}$. To limit the lattice matching, we need a $(6\sqrt{2} \times 6\sqrt{2})$ cell ($25.596 \text{ \AA} \times 25.596 \text{ \AA}$) for CoO (100), and a (8×4) cell ($24.984 \text{ \AA} \times 25.580 \text{ \AA}$) for RuO₂ (110) to build an interface. There are 144 Co and 144 O atoms for one layer CoO (100), and 128 O and 64 Ru atoms for one layer RuO₂ (110). To characterize reactions on interface, the super-lattice slab models need to take at least three layers for each material. Therefore, there would be 1440 atoms (432 Co,

816 O, and 192 Ru) for the interface model, which is too large to compute. While the experimental data already show the edge of the interface dominates the reaction (Please *see* our response to your comment 1-1), and RuO₂ is cluster. Consequently, a nanoscale cluster was used to represent RuO₂ in the calculations.

Response to Reviewer #2

General Comments:

The electrocatalytic splitting of water has drawn great interest these years. This paper reports the RuO₂/CoO_x hybrid catalyst to break the stability and activity limits of RuO₂ for stable and efficient OER catalyst in neutral and alkaline media. The sacrificial oxidization of CoO_x and the electron interaction among the face-to-face Ru–O–Co interfacial atoms enhance the stability, while the Ru/Co dual-atom site exposed around the interface is responsible for the improved activity. The authors have delivered detailed characterizations and thorough analyses of those catalysts, which provides an atomic-scale understanding of the interfacial effect of Ru–O–Co sites for simultaneously enhancing the stability and activity of RuO₂. Overall, the paper is well written and the experiments are carefully performed.

Response:

We would like to thank the Reviewer for his/her valuable comments and positive recommendation.

General Comments:

However, I think the total novelty of this work is not enough, as a similar approach has been published by Yu et al. (ACS Sustainable Chem. Eng. 2020, 8, 17520–17526) using an amorphous CoO_x decorated crystalline RuO₂ nanosheet catalyst, which is also a highly active and stable OER catalyst under alkaline condition. The author should explain the novelty of their approach as compared to Yu et al. We will reconsider this draft after addressing all comments.

Response:

Thanks for your valuable comment. Yu *et al.* reported a 48 h durability of CoO_x–RuO₂, and claimed that “After the OER durability test, the shape of Ru 3p peaks remains the same, but ***the intensity reduces, suggesting the loss of the Ru element, which is instinctive after the durability test.***³⁴” Therefore, the electrochemical corrosion of RuO₂ has not been solved in their work. Besides,

the authors simply ascribed the improved activity of CoO_x-RuO₂ (0.26 V overpotential @ 10 mA cm⁻² in 1.0 M KOH) to sufficient active sites and binder free of CoO_x-RuO₂. In our work, we tackle the electrochemical corrosion of RuO₂ under OER conditions and achieve record high OER activities under both neutral and alkaline conditions. Moreover, the physical origin of enhanced stability and activity of RuO₂ is clearly shown in our work. We note that although their material system is similar with ours, the amorphous structure of CoO_x limits the strong atomic and electronic interaction between CoO_x substrate and RuO₂, which is however the novelty of our work.

Perhaps in the original submission we failed to underline the significance and novelty of this work. Therefore, we would like to highlight them below as compared to the key points of the works of Yu *et al.* and other related works:

1. This work represents the first attempt to exceed the Pourbaix stability limit of bulk RuO₂ under OER conditions. Although some works have reported enhancing the stability of Ru-based catalysts, there still lacks a general strategy that can fundamentally suppress the electrochemical corrosion of RuO₂ under OER conditions. We propose to stabilize RuO₂ via constructing stable RuO₂/CoO_x interface. Specifically, the preferential oxidation of CoO_x during OER process and the electron gain of RuO₂ through the RuO₂/CoO_x interface remarkably reduce the driving force of RuO₂ dissolution in the hybrid, which makes the stability of ~2 nm RuO₂ nonoclusters in the hybrid far beyond the bulk Pourbaix limits of RuO₂ (Fig. 1a, b). As a result, RuO₂/CoO_x hybrid exhibits an outstanding stability for more than 200 h, withstanding anodic potential as high as 2 V_{RHE} with no Ru dissolution being detected, as verified by *in situ* X-ray photoelectron spectroscopy and inductively coupled plasma mass spectrometry.

2. This work achieves a record high OER activity. The work of Nørskov *et al.* (*J. Phys. Chem. C* 2017, 121, 18516) has demonstrated that the experimentally observed high activity of RuO₂ is derived from the unstable high-valence Ru species while the activity of the ‘stable’ RuO₂ is unsatisfactory. We demonstrate that the highly active interfacial Ru/Co dual-atom sites can synergistically adsorb and stabilize the key *OOH intermediate during OER, demonstrating a distinct catalytic adsorption behaviour from the pristine RuO₂. This transforms the rate-determining step of OER from *OOH formation to the subsequent *O₂ desorption step with considerably decreased potential and energy barrier, which was verified by isotope labelling, *in situ* infrared reflection and theoretical calculations. As a result, RuO₂/CoO_x hybrid affords an ultra-low overpotential of 0.16 V to drive an OER current of 10 mA cm⁻² in 1.0 M KOH.

3. We attempt to break the stability and activity limits of RuO₂ by geometrically and electronically decoupling its stability-activity relation. Although some works have reported Ru based electrocatalysts with both enhanced stability and activity, they usually lumped the origin of enhanced activity and stability together, such as changes in the electronic structure of Ru sites (*Nat.*

Catal. 2019, 2, 304; *Nat. Commun.* 2019, 10, 1711). The atomic-scale understanding of the decoupled stability and activity in one catalyst system is still lacking. We attempt to break the stability and activity limits of RuO₂ by geometrically and electronically decoupling its stability-activity relation through constructing the RuO₂/CoO_x interface, where sacrificial oxidization of CoO_x substrate and electron interaction among the face-to-face Ru-O-Co interfacial atoms enhance the stability, while Ru/Co dual-atom site exposed around the interface is responsible for the improved activity.

Accordingly, to address this comment, we have highlighted the novelty of this work **from line 18, page 1 to line 1, page 2** in the revised Abstract,

“It remains a grand challenge to achieve stable and active RuO₂ electrocatalyst as the current synthetic strategies usually enhance one of the two properties at the expense of the other. Here, we report breaking the stability and activity limits of RuO₂ by artificially constructing a RuO₂/CoO_x interface. We demonstrate that RuO₂ can be greatly stabilized on the CoO_x substrate to exceed the Pourbaix stability limit of bulk RuO₂. This is realized by the preferential oxidation of CoO_x during OER process and the electron gain of RuO₂ through the interface. Besides, a highly active Ru/Co dual-atom site can be generated around the RuO₂/CoO_x interface to synergistically adsorb the oxygen intermediates, leading to a favourable reaction path and record high OER activities under both neutral and alkaline conditions.”

and in **lines 14-16, page 9**, in the revised conclusion.

“With such unique electronic and geometric effects generated by the RuO₂/CoO_x interface, we solved the critical issues of RuO₂ under OER conditions and achieved high stability and outstanding activity.”

Moreover, we have cited the reference the Reviewer mentioned as **Reference [19]** in Introduction and in the Reference list.

Original comment 2-1:

Page 4 “Note that the change in the Co charge at the interface relative to the value in the bulk CoOOH is negligible (Fig. 1f).” Why? The difference of Co bader charge between bulk and interface can be negligible?

Response:

Sorry for unclear expression. We meant that the change of Co charge at the interface relative to the value in CoOOH is ~ 0 ($\Delta Q_{\text{Co}} = Q_{\text{Co}}^{\text{interface}} - Q_{\text{Co}}^{\text{CoOOH}}$). Accordingly, to avoid misunderstanding, the related description has been revised **in lines 11-12, page 4** in the revised manuscript.

“Note that the Co charge at the interface is almost identical to that in the bulk CoOOH (Fig. 1f).”

Original comment 2-2:

Authors mentioned that “Once Ru–O–Co bond is formed at the RuO₂/CoO_x interface, the electron-

rich O ions connecting with Co ions contribute electrons to the nearby Ru ions through metal-oxygen re-hybridization, thus enriching electrons in the interfacial Ru sites". The Ru-O-Co bond only improves the stability of interfacial RuO₂, how this effect can be extended into the dominated RuO₂ surface (surface provides more active sites towards OER) since the limited interfacial bonding. Not only are the interfacial species active sites, but outer surface RuO₂ also. How can you distinguish those?

Response:

Thanks for your valuable comment. Our EELS measurement (Fig. 2e-f) shows that the enrichment of Ru charge at the interface will affect the distribution of Ru charge in the bulk and on the surface of RuO₂ through continuous Ru-O bonds. The smaller the RuO₂ particle size is, the greater the interface effect will be. In our case, the size of RuO₂ nanoparticles deposited on CoO_x is only 2 nm, which falls within the range of strong interfacial interaction (*Chem. Rev.* 2018, 118, 4981). This result is supported by the calculated Pourbaix diagram of RuO₂/CoO_x (Fig. 1b) that in the OER potential range, the entire RuO₂ particle (including the surface) is stabilized on CoO_x.

Regarding the active sites, we carefully performed a series of control experiments to verify that the interfacial Ru/Co dual-atom sites are the most active sites for OER. Please *see* details in Response to your comment 2-5. It is interesting to find that although the number of interfacial atoms is relatively small compared with that of surface Ru and Co sites on pristine RuO₂ and CoO_x, respectively, the OER mechanism on these interfacial sites is different from that on pristine RuO₂ and CoO_x. This is the origin of the significantly enhanced performance of RuO₂/CoO_x.

In response, we have added related discussion in **lines 20-21, page 5** in the revised manuscript. *"We note that the enrichment of Ru charge at the interface will affect the distribution of Ru charge in the bulk and on the surface through continuous Ru-O bonds."*

Original comment 2-3:

Authors roughly conclude that the EELS analysis supports the calculated charge changes, especially O ions from CoO_x to RuO₂ via the interface, please clearly describe.

Response:

According to the suggestion of the Reviewer, the related description has been revised in **lines 9-21, page 5** in the revised manuscript.

"As illustrated in Fig. 2e, the collected Ru-M_{2,3} spectrum at the interface (point 3) shifts 0.3 eV toward the low energy loss direction with respect to that of RuO₂ (point 1), indicating a decreased Ru valence at the interface. For O-K edge spectra (Fig. 2f), the curves show obvious shape change from

RuO₂-like (point 1) to CoO_x-like (point 5). In particular, the characteristic peak ‘a’ collected in CoO_x gradually weakens towards the interface until disappears in RuO₂. This reflects different electronic properties of O atoms connecting with Ru and Co atoms, respectively, and re-hybridization of O atoms at the interface caused by simultaneous connection with Ru and Co atoms. Notably, no noticeable peak shift is observed in the collected Co-L_{2,3} spectra (Fig. 2g and Supplementary Fig. 9). These experimental results well support the calculated evident charge change of O ions from CoO_x to RuO₂ via the interface (Fig. 1e), while no significant Co charge change from bulk CoO_x to the interface (Fig. 1f). This indicates that O ions play a decisive role in the reduction of Ru valence through the electronic interaction among Ru, O and Co atoms at the interface. We note that the enrichment of Ru charge at the interface will affect the distribution of Ru charge in the bulk and on the surface through continuous Ru-O bonds.”

Original comment 2-4:

From the in situ IR spectroscopy, the O-O bond formation can be confirmed as RDS on RuO₂, and a different OER mechanism occurs on RuO₂/CoO_x, what exact mechanism is? Please emphasize here.

Response:

Thanks for your comment. We meant the different OER mechanism is the different RDS of OER. The generally accepted OER steps (*Chem. Soc. Rev.* 2017, 46, 337) are

where * is the active site. As discussed the main text, for RuO₂/CoO_x, O-O bond formation (equation 3) is not the RDS. Our calculation (Fig. 5c) reveals the RDS of RuO₂/CoO_x transfers to subsequent step of *OOH formation – that is, desorption of O₂ (equation 4).

In response, we have clarified the related discussions on the RDS of RuO₂/CoO_x **in lines 23-24, page 7,**

“These results suggest that the RuO₂/CoO_x exhibits a different RDS compared with the pristine RuO₂ as we will discuss in detail later.”

and **in lines 3-5, page 9** in the revised manuscript.

*“More importantly, this shifts the RDS of RuO₂/CoO_x to the subsequent step of *OOH formation – that is, desorption of *O₂ (Supplementary Note 4), which demonstrates a significantly decreased energy injection of 0.50 eV (Fig. 5c).”*

Moreover, **Supplementary Note 4** was added in the revised supporting information to discuss the

RDS of OER.

“Supplementary Note 4. OER RDS of RuO₂/CoO_x.

Notably, the generally accepted reaction steps⁴ for OER under neutral conditions are

where * is the active site. As discussed the main text, for RuO₂/CoO_x, O–O bond formation (equation 13) is not the RDS. Our further calculation (Fig. 5c) reveals the RDS of RuO₂/CoO_x transfers to subsequent step of *OOH formation – that is, desorption of *O₂ (equation 14).”

Original comment 2-5:

Since the Ru-Co dual-atoms sites are served as co-works active sites in simulations analysis, the evidence of the corresponding experiments discovery cannot be missed.

Response:

Thanks for your kind suggestion. We have performed additional experiments to verify Ru/Co dual-atom sites as the most active OER sites. First, we deposited RuO₂ nanoparticles on carbon black with the same particle size (~2 nm) as that on RuO₂/CoO_x (Fig. R3a and R3b). The much higher current density of RuO₂/CoO_x compared with that of RuO₂ with the same RuO₂-mass (Fig. R3c) indicates the crucial role of the interface in enhancing OER performance. Moreover, we prepared RuO₂/CoO_x with varied RuO₂ particle size or loading masses (Fig. R4a). Assuming either surface Ru site or interfacial Ru-Co dual-atom site as the most active sites, the corresponding site number was calculated and correlated with the OER current density at 1.50 V_{RHE} (Fig. R4b and R4c). An adequate linear relationship was observed between the number of interfacial Ru-Co dual-atom site and the OER current density (Fig. R4c). These collective results reveals that the interfacial Ru/Co dual-atoms are the most active OER sites.

Figure R3. (a) TEM image of RuO₂ deposited on carbon black. (b) Polarization curves of the RuO₂

with different RuO₂ mass loadings on per cm² electrode in neutral electrolyte. (c) Comparison of current density (J) of RuO₂/CoO_x and RuO₂ at 1.50 V_{RHE} with different RuO₂-masses.

Figure R4. (a) Polarization curves of the RuO₂/CoO_x with varied RuO₂ particle size or loading masses in neutral electrolyte. (b) and (c) Plot of current density (J) for RuO₂/CoO_x at 1.50 V_{RHE} as a function of the estimated number of surface Ru sites and interfacial Ru/Co dual-atom sites, respectively.

In response, we have added Figs. R3 and R4 as **Supplementary Figs. 34 and 36** in the revised supporting information. Moreover, **Supplementary Fig. 35** was added to show the characterization of RuO₂/CoO_x with different RuO₂ particle sizes. In addition, **Supplementary Note 4** was added to show details of Experiments that validated Ru/Co dual site as the active site for RuO₂/CoO_x.

“Supplementary Note 4. Experimental validation of Ru/Co dual site as the active site for RuO₂/CoO_x.”

First, we deposited RuO₂ nanoparticles on carbon black with the same particle size (~2 nm) as that on RuO₂/CoO_x (Supplementary Fig. 19c). As shown in Supplementary Fig. 34b, RuO₂/CoO_x exhibits a much higher current density than RuO₂ with the same RuO₂-mass loading, indicating the crucial role of interface in enhancing OER performance. Second, we prepared RuO₂/CoO_x with varied particle sizes (Supplementary Fig. 35) or loading masses of RuO₂. Assuming either surface Ru site or interfacial Ru-Co dual-atom site as the most active sites, the corresponding site number can be calculated. Specifically, the number of interfacial Ru/Co dual-atom sites ($N_{\text{interfacial Ru/Co dual-atom site}}$) can be numerically calculated by

$$N_{\text{interfacial Ru/Co dual-atom site}} = N_{\text{RuO}_2} \times n_{\text{interfacial Ru/Co dual-atom site}} \quad (16)$$

where N_{RuO_2} is the total number of RuO₂ nanoparticles deposited on CoO_x, and $n_{\text{interfacial Ru/Co dual-atom site}}$ is the number of interfacial Ru/Co dual-atom sites on each RuO₂ particle. Assuming that the RuO₂ nanoparticles are cube-shaped with an average side length (a) of 2 nm, N_{RuO_2} can be calculated by

$$N_{\text{RuO}_2} = \frac{m_{\text{RuO}_2}}{a^3 \times \rho} \quad (17)$$

where m_{RuO_2} is the loaded RuO_2 -mass and ρ is the density of RuO_2 . Moreover, $n_{\text{interfacial Ru/Co dual-atom site}}$ can be obtained by

$$n_{\text{interfacial Ru/Co dual-atom site}} = 4 \times (a/r_{\text{Ru-Ru}}) \quad (18)$$

where $r_{\text{Ru-Ru}}$ is the nearest neighboring Ru-Ru distance measured in supplementary Fig. 15. The number of surface Ru sites ($N_{\text{surface Ru site}}$) can be numerically calculated by

$$N_{\text{surface Ru site}} = N_{\text{RuO}_2} \times n_{\text{surface Ru site}} \quad (19)$$

where $n_{\text{surface Ru site}}$ is the number of surface of Ru sites on each RuO_2 particle, which can be calculated by

$$n_{\text{interfacial Ru/Co dual-atom site}} = 5 \times (a/r_{\text{Ru-Ru}})^2 \quad (20)$$

We correlated $N_{\text{interfacial Ru/Co dual-atom site}}$ and $N_{\text{surface Ru site}}$ with the OER current density of these samples at 1.50 V_{RHE} (Supplementary Fig. 36). An adequate linear relationship was observed between $N_{\text{Ru/Co dual-atom sites}}$ and the OER current density (Supplementary Fig. 36c). Therefore, these collective results reveals that the interfacial Ru/Co dual-atoms are the most active OER sites.”

Original comment 2-6:

Page 8 “Relative shift of *OOH bands is observed in the in situ IR spectra of the $\text{RuO}_2/\text{CoO}_x$ compared with those of RuO_2 (Fig. 4e, f),” Please mark the indicated bands in the corresponding figures.

Response:

Thanks for your kind suggestion. We have marked the indicated bands in the revised **Fig. 4e, f**.

Figure R5. *In situ* surface-enhanced IR spectra of RuO₂/CoO_x and RuO₂ at different potentials.

Original comment 2-7:

“..., verifying the adsorption configuration of *OO–H...O, which facilitates the stabilization of *OOH at the Ru/Co dual-atom site around the interface (inset of Fig. 5c and Supplementary Fig. 27).”
As mentioned above, how to experimentally prove these configurations on Ru-Co dual-sites?

Response:

Thanks for your comment. According to previous work (*J. Phys. Chem. A* 2018, 122, 4481), in IR spectra, when intermolecular hydrogen bond is formed and stretches the target bond in the probe molecule, the vibrational frequency of the probe molecule will shift toward the low wavenumber direction. Relative*OOH band shift is observed in the *in situ* IR spectra of the RuO₂/CoO_x compared with those of RuO₂ (Fig. 4e, f), verifying the adsorption configuration of *OO–H•••O.

In response, we have added related discussion **in lines 15-20, page 8** in the revised manuscript. “According to previous work³⁵, when the intermolecular hydrogen bond stretches the bond in the probe molecule, it will lead to a shift of the stretching vibrational frequency of the probe groups toward the low wavenumber direction in IR spectra. Relative shift of *OOH bands is observed in the *in situ* IR spectra of the RuO₂/CoO_x compared with those of RuO₂ (Fig. 4e, f), verifying the adsorption configuration of *OO–H...O, which facilitates the stabilization of *OOH at the Ru/Co dual-atom site around the interface (inset of Fig. 5c and Supplementary Fig. 32).”

Original comment 2-8:

We believe that the Ru-Co dual sites practically work as synergistic catalysis, but the portion of interfacial atoms are limited what about else Ru species and substrate CoO_x ? Do those non-interfacial species also catalyze OER? Some control experiments are missed.

Response:

Thanks for your kind suggestion. The activity of CoO_x is much lower than that of $\text{RuO}_2/\text{CoO}_x$ (Fig. 4a). We compared the performance of RuO_2 and $\text{RuO}_2/\text{CoO}_x$ with the same RuO_2 particle size and RuO_2 -mass. Please see details in Response to comment 2-5. These collective results confirm the Ru/Co dual-atom site as the most active site for OER. We would like to say that although the number of interfacial atoms is limited compared with that of surface Ru and Co, the OER mechanism on these interfacial sites is different from that on pristine RuO_2 and CoO_x . This is the origin of the significantly enhanced performance of $\text{RuO}_2/\text{CoO}_x$.

Original comment 2-9:

The information about Ru-O-Co bonds should be additionally characterized and provided.

Response:

Thanks for your kind suggestion. Accordingly, we have characterized the Ru-O-Co bonds by Fourier transform extended X-ray absorption fine structure (FT-EXAFS). As shown in Fig. R6a, a new peak assigned to Ru-O-Co bonds (with shorter Ru-O distance than that in RuO_2) appears in the FT-EXAFS spectrum of $\text{RuO}_2/\text{CoO}_x$. In response, we have added Fig. R6 as **Supplementary Fig. 10** in the revised supporting information and added the above related discussion in the **caption of Supplementary Fig. 10**.

Figure R6. (a) FT-EXAFS spectra of $\text{RuO}_2/\text{CoO}_x$, RuO_2 and Ru foil. (b) Schematic diagram showing the Ru-O bonds in bulk RuO_2 and in interfacial Ru-O-Co.

Response to Reviewer #3

General Comments:

Manuscript by Du et al. describes the synthesis of RuO₂ nanoparticles deposited on bulk CoO_x support and application of this material for the alkaline oxygen evolution reaction. The study discusses an interesting concept and is generally carefully done, therefore can be recommended for publication in Nature Communications once the following issues are addressed.

Response:

We would like to thank the Reviewer for his/her valuable comments to help us to improve the quality of this manuscript.

Original comment 3-1:

Intrinsic activity of the materials (normalized to the BET/electrochemically active surface area) rather than specific activity should be compared to prove that the electronic effects of the CoO_x support rather than enhanced dispersion of the active species (compared to bulk RuO₂) are responsible for the activity improvement (RuO₂/CoO_x vs. CoO_x).

Response:

Thanks for your kind suggestion. Accordingly, we have tested the electrochemically active surface areas (ECSAs) of RuO₂/CoO_x, RuO₂ and CoO_x (Fig. R7a-d), and normalized their OER current density to the corresponding ECSA to obtain the specific current density (J_{ECSA}). As shown in Fig. R7e, J_{ECSA} of RuO₂/CoO_x is higher than those of RuO₂ and CoO_x, demonstrating that the electronic effect between RuO₂ and CoO_x is indeed the reason for the enhanced activity of RuO₂/CoO_x.

Figure R7. (a-c) Plots of current density (J) versus scan rate for $\text{RuO}_2/\text{CoO}_x$, RuO_2 and CoO_x , respectively. (d) Determined double-layer capacitance (C_{dl}) for $\text{RuO}_2/\text{CoO}_x$, RuO_2 and CoO_x . Note that the electrochemically active surface area (ECSA) of the catalyst can be calculated by $\text{ECSA} = \frac{C_{dl}}{C_s} \times S_{\text{geometric}}$, where C_s is $60 \mu\text{F}\cdot\text{cm}^{-2}$, $S_{\text{geometric}}$ is the geometric area of the glassy-carbon electrode, and C_{dl} was the measured electrical double-layer capacitor of the catalyst. (e) Intrinsic activity of $\text{RuO}_2/\text{CoO}_x$, RuO_2 and CoO_x . Note that J_{ECSA} is obtained by normalizing the OER current density to the calculated ECSA.

In response, we have added Fig. R7 as **Supplementary Fig. 27** in the revised supporting information.

Original comment 3-2:

Turnover frequency (P. 7) cannot be calculated based on Ru content only as there are no strong evidences that Ru is the only reaction site.

Response:

Thanks for your valuable comment. We agree with the Reviewer that the TOF should be calculated based on the number of the “real” reaction sites. In our case, the real active sites for $\text{RuO}_2/\text{CoO}_x$ are the interfacial Ru/Co dual-atom sites. To verify this point, we have performed additional experiments. First, we deposited RuO_2 nanoparticles on carbon black with the same particle size (~ 2 nm) as that

on RuO₂/CoO_x (Fig. R8a and R8b). The higher current density of RuO₂/CoO_x compared with that of RuO₂ with the same RuO₂-mass (Fig. R8c) indicates the crucial role of the interface in enhancing OER performance of RuO₂/CoO_x. Moreover, we prepared RuO₂/CoO_x with varied RuO₂ particle size or loading masses (Fig. R9a). Assuming either surface Ru site or interfacial Ru-Co dual-atom site as the most active sites, the corresponding site number was calculated and correlated with the OER current density at 1.50 V_{RHE} (Fig. R9b and R9c). An adequate linear relationship was observed between the number of interfacial Ru-Co dual-atom sites and the OER current density (Fig. R9c). These collective results reveals that the interfacial Ru/Co dual-atoms are the most active OER sites.

In the original submission, we calculated the TOF by normalizing the production rate of O₂ to the total Ru atoms on RuO₂/CoO_x. It is an underestimation, however it is generally used in literature (*Nat. Energy* 2016, 1, 16053). Therefore, in order to compare our data to those in the literature, we calculated the TOF of RuO₂/CoO_x based on the total Ru atoms. According to the comment of the Reviewer, we have added related discussion in **Supplementary Note 3** in the revised supporting information.

“Note that the true TOF of the most active Ru/Co dual-atom sites (Supplementary Figs. 34 and 36) should be even higher because the number of Ru/Co dual-atom sites is much smaller than the total number of Ru sites on RuO₂/CoO_x.”

Figure R8. (a) TEM image of RuO₂ deposited on carbon black. (b) Polarization curves of the RuO₂ with different RuO₂ mass loadings on per cm² electrode in neutral electrolyte. (c) Comparison of current density (J) of RuO₂/CoO_x and RuO₂ at 1.50 V_{RHE} with different RuO₂-masses.

Figure R9. (a) Polarization curves of the RuO₂/CoO_x with varied RuO₂ particle size or loading masses in neutral electrolyte. (b) and (c) Plot of current density (J) for RuO₂/CoO_x at 1.50 V_{RHE} as a function of the estimated number of surface Ru sites and interfacial Ru/Co dual-atom sites, respectively.

Original comment 3-3:

Citation [30] does not refer to the study describing ¹⁶O/¹⁸O isotope effect (P. 7).

Response:

Sorry for listing the reference carelessly. We have replaced reference [31] (original reference [30]) in the revised reference list.

Original comment 3-4:

I recommend to refrain from using phrases like “innovatively constructed”, “dream RuO₂”, “unprecedented high stability” to highlight the properties of the material.

Response:

Thanks for your kind suggestion. The related descriptions mentioned by the Reviewer have been revised in the revised manuscript.

Original comment 3-5:

Fig. 1b: Ru species are missing from the low potential/pH region; Fig. 3f: what are the potentials applied?

Response:

Thanks for your kind noticing. “RuO₂” has been added in the low potential/pH region in the revised **Fig. 1b**, and the applied potentials were indicated in the **caption of Fig. 3f**.

Response to Reviewer #4

General Comments:

The manuscript describes the synthesis of a new RuO₂-based electrocatalyst consisting of RuO₂/CoO_x hybrid for the OER with high activities in both neutral and alkaline conditions. This paper is based on the suppression of the electrochemical corrosion of Ru species by fabricating a RuO₂/CoO_x hybrid to limit the dissolution of Ru species through the oxidation of Co-based species which are more stable. The paper presents a lot of sophisticated techniques combined with DFT to account for their claims but, in my opinion, the materials characterizations are not very convincing to account for the theoretical description of it. As a consequence, the theoretical results are very interesting but it seems they may not reflect the actual behavior of these materials. Moreover, as stated below, no mention of the acidic conditions are made where this strategy would really have been an asset to stabilize Ru-based catalysts so they could substitute Ir-based electrocatalysts. In general, many questions arise from the experimental part and how the authors draw conclusions from the experimental data. My opinion is that the paper should be revised prior to publication and maybe in a more catalysis-oriented journal.

Response:

We would like to thank the Reviewer for his/her valuable comments to help us to improve the quality of this manuscript. We have added additional experimental evidences and descriptions of our samples in the revised manuscript and supporting information. We believe that our strategy proposed in this work is applicable to RuO₂ in acidic solution and other highly active materials with unsatisfactory stability if proper substrate is selected. Moreover, we confirm that the structure of our RuO₂/CoO_x is consistent with the computational model, and the conclusion drawn in this work is trustful. Please *see* details in our responses to your comments.

Original comment 4-1:

The use of CoO_x is somehow confusing: is it CoO with vacancies, does it represent the CoO precursor being oxidized into different others structure with different oxidation number for Co? For instance in the caption of figure 2: is it CoO or CoO_x?

Response:

Thanks for your kind notice. CoO_x represents the CoO precursor being oxidized into different

cobalt oxides with different Co valent states. In the caption of Fig. 2, the support to load Ru nanoparticles is CoO. Accordingly, we have clarified the definition of CoO_x **in lines 20-22, page 3** in the revised manuscript.

“The CoO_x support is gradually oxidized from CoO to Co₃O₄, CoOOH and eventually CoO₂ with the increase of anodic potential. Hereafter, CoO_x repents these cobalt oxides for simplicity.”

Moreover, we have carefully checked all the descriptions of “CoO_x” and “CoO” in the revised manuscript and supporting information.

Original comment 4-2:

How the value of the concentration for the Pourbaix diagram (10⁻⁶ M) was chosen and why is it similar for the calculation of Co-based, Ru-based and Co/Ru-based materials?

Response:

Thanks for your comment. 10⁻⁶ M is a typical concentration for Pourbaix diagram calculation (*Nat. Energy* 2017, 2, 17070; *Phys. Rev. B* 2012, 85, 235438), which represents a considerable dissolution. Therefore, it was chosen for calculation of the three kinds of materials in this work.

Original comment 4-3:

How is defined the formation energy of RuO₂/CoO_x hybrids (equation and free Gibbs energy definition)?

Response:

Thanks for your comment. Accordingly, we have clarified the related definition of the formation energy in the **caption of Supplementary Fig. 2**.

“Supplementary Figure 2. Theoretical investigation on the adhesion energy of the RuO₂ cluster on CoO_x substrates. (a) RuO₂/Co₃O₄. (b) RuO₂/CoOOH. (c) RuO₂/CoO₂. Note that the adhesion energy (ΔE) was calculated by

$$\Delta E = E_{\text{RuO}_2/\text{CoO}_x} - E_{\text{RuO}_2} - E_{\text{CoO}_x} \quad (21)$$

where $E_{\text{RuO}_2/\text{CoO}_x}$, E_{RuO_2} , and E_{CoO_x} are the calculated energies for RuO₂/CoO_x, RuO₂ and CoO_x, respectively. The negative values of ΔE indicate a strong adhesion of RuO₂ to the CoO_x support. This undoubtedly decreases the driving force for RuO₂ dissolution, thus stabilizes RuO₂ in the hybrid catalyst.”

Original comment 4-4:

Typo: On the SI, supplementary note 2. Estimation of the proportion.

Response:

Thanks for your kind notice. We have revised the related typo mistake in the revised supporting information.

Original comment 4-5:

The authors mentioned two types of substrates: Carbon fiber paper and FTO. Does the choice of the substrate has an influence on the growth of the CoO_x rods and the morphology of the samples? References 10.1038/ncomms12876 and 10.1038/s41467-017-01872-y only mention carbon fiber paper, stainless steel and carbon nanotube films and one can see differences in the density of the forest of rods and maybe length and diameter of the rods?

Response:

Thanks for your comment. The diameter and density of CoO nanorods are controlled by the structure of pre-synthesized ZnO nanorods, which were used as the sacrificial templates for CoO nanorods. As shown in Fig. R10, CoO nanorods grown on carbon fiber substrate (CFP) and FTO exhibit almost identical morphology. In this work, in order to compare the performance of $\text{RuO}_2/\text{CoO}_x$ with that of pristine RuO_2 (grown on carbon black) and commercial RuO_2 , we need to strip the $\text{RuO}_2/\text{CoO}_x$ from the substrate to prepare the catalyst ink for OER performance testing. Therefore, FTO substrate with smooth surface (easy for $\text{RuO}_2/\text{CoO}_x$ being peeled off) was used.

Figure R10. (a) and (b) SEM images of CoO nanorods grown on CFP and FTO, respectively.

Original comment 4-6:

We can see some darker circular stains on the CoO_x nanorods (fig 2b, fig. S14a), from where does it come?

Response:

Thanks for your comment. The darker circular stains mentioned by the Reviewer are nanopores (NPs) generated by the release of lattice strain between the template (ZnO) and product (CoO) during the cation exchange process. It is a common structural characteristic in materials prepared by cation exchange method (*Chem. Soc. Rev.* 2013, 42, 89). Fig. R11 is a TEM image of the nanorod, and the nanopores are indicated.

Figure R11. TEM image of as-exchanged CoO nanorod.

In response, we have indicated the nanopores in **Supplementary Fig. 17b** (original Supplementary Fig. 14a) and added related discussion and the above reference in the **caption of Supplementary Fig. 17**.

Original comment 4-7:

Fig S4: what is the substrate (FTO or CFP)?

Response:

Thanks for your comment. In previous XRD test, CoO nanorods were stripped from FTO substrate and loaded on CFP for characterization (the signal of the FTO substrate is too strong). According to the comment 4-5 raised by the Reviewer, we realized that it may be a little misleading. Therefore, we have re-tested XRD spectra of CoO and Ru/CoO without any substrate (Fig. R12).

Accordingly, we have replaced the XRD spectra in **Supplementary Fig. 3** with the newly tested ones.

Figure R12. X-ray diffraction (XRD) patterns of CoO and Ru/CoO.

Original comment 4-8:

Fig S7 shows the electrochemical oxidation of Ru/CoO_x between 0.8 V and 1.5 V vs RHE while it is said to be between 1.0 and 1.5 V in the experimental part? Which one is the good value? It is the same range for RuO₂ only? It seems that in one case, the potential is set to a value where the catalytic regime occurs while in the other case no (cf fig. 4a), does it have an impact on the catalysis afterwards?

Response:

It is a careless typing, which causes misleading. In fact, the potential range (0.8~1.5 V_{RHE}) is applied to ensure complete transformation of Ru to RuO₂ and a narrower potential has no significant impact on the catalytic activity (Fig. R13).

Figure R13. The effect of potential range on catalytic activity of RuO₂/CoO_x. (a) CV cycling curves of the electrochemical oxidization of Ru/CoO to form the RuO₂/CoO_x. (b) OER polarization curves of RuO₂/CoO_x formed in (a).

Thank you again for finding and pointing our mistake. Accordingly, we have revised the related description in the experimental part.

“Finally, the obtained Ru/CoO nanorods were electrochemically oxidized by scanning cyclic voltammetry between 0.80~1.50 V_{RHE} to attain RuO₂/CoO_x catalysts.”

Original comment 4-9:

It is possible to compare two systems which do not have the same morphology (RuO₂ cf. fig 21 and RuO₂/CoO_x, cf fig 2) but the authors should give some impacts of the morphology on the electrochemical activity and compare data for a same given mass or given surface area since the RuO₂ reference is not a commercial sample.

Response:

Thanks for your valuable comment. We completely agree with the Reviewer that the morphology has an important impact on electrochemical activity of catalysts. Therefore, the size and distribution of RuO₂ deposited on carbon black was controlled as identical to those on CoO_x (Fig. R14). Consequently, the current activity difference between RuO₂/CoO_x and RuO₂ can explain the role of RuO₂/CoO_x interface. Moreover, according to the suggestion of the Reviewer, we have compared the OER performance of RuO₂/CoO_x and RuO₂ with identical RuO₂-mass loading. As shown in Fig. R15c, RuO₂/CoO_x exhibits a much higher current density than RuO₂ with the same RuO₂-mass loading.

In response, we have added Figs. R14 and R15 in **Supplementary Figs. 17, 19, 25, and 34** in the revised supporting information.

Figure R14. (a1, a2) and (b1, b2) TEM images of RuO₂/CoO_x and RuO₂ deposited on carbon black with identical RuO₂-mass, respectively.

Figure R15. (a) and (b) OER polarization curves of RuO₂/CoO_x and RuO₂ with different RuO₂-masses on per cm² electrode, respectively. (c) Comparison of current densities of RuO₂/CoO_x and RuO₂ at 1.5 V_{RHE} with identical RuO₂-masses.

Original comment 4-10:

Supplementary tables 3 and 4: I do understand how the RuO₂ wt% is calculated thanks to the footnote a but then the RuO₂ wt% in table 4 is calculated from the mass of the CFP support so why the calculation on table 3 does not include the support? Why do the authors use a wt% and not an atomic

percentage?

Response:

Thanks for your comment. For OER tests, RuO₂/CoO_x (peeled from substrate) and RuO₂ (deposited on carbon black) were mixed with additional carbon black to prepare the catalyst ink. To avoid misleading, we have listed the loading masses of all the components used for OER tests in the tables. Moreover, atomic percentage was used according to the suggestion of the Reviewer.

Table R1. Details of preparing RuO₂/CoO_x^a for OER test.

Catalyst	m_{RuO_2} ($\mu\text{g cm}^{-2}$) ^b	m_{CoO_x} (mg cm^{-2}) ^c	Ru (at%) ^d	$m_{\text{Carbon black-ink}}$ (mg cm^{-2}) ^e	Total mass (mg cm^{-2}) ^f
1	3	0.25	0.7	0.25	0.50
2	6	0.25	1.3	0.25	0.50
3	10	0.25	2.1	0.25	0.50
4	17	0.24	3.9	0.25	0.50
5	30	0.23	7.0	0.25	0.50

^aloaded 0.05 mg RuO₂/CoO_x and 0.05 mg carbon black on RDE (0.196 cm²). ^bmass of RuO₂ on per cm² RDE, measured by ICP-MS. ^cmass of CoO_x on per cm² RDE, $m_{\text{CoO}_x} = (0.05 \text{ mg} - m_{\text{RuO}_2}) / 0.196 \text{ cm}^2$. ^dRu (at%) = $N_{\text{Ru}} / (N_{\text{Ru}} + N_{\text{Co}})$, where N_{Ru} and N_{Co} are the number of Ru and Co in the RuO₂/CoO_x catalyst, respectively. ^eadded in the catalyst ink. ^fTotal mass = $m_{\text{RuO}_2} + m_{\text{CoO}_x} + m_{\text{Carbon black}}$.

Table R2. Details of preparing reference RuO₂^a (deposited on carbon black) for OER test.

Catalyst	m_{RuO_2} ($\mu\text{g cm}^{-2}$) ^b	$m_{\text{Carbon black}}$ (mg cm^{-2}) ^c	$m_{\text{Carbon black-ink}}$ (mg cm^{-2}) ^d	Total mass (mg cm^{-2}) ^f
1	23	0.23	0.26	0.50
2	50	0.20	0.26	0.50
3	84	0.17	0.26	0.50
4	126	0.13	0.26	0.50

^aloaded 0.05 mg RuO₂ (with supporting carbon black) and 0.05 mg carbon black on RDE (0.196 cm²). ^bmass of RuO₂ on per cm² RDE, measured by ICP-MS. ^cmass of carbon black for depositing RuO₂ on per cm² RDE, $m_{\text{Carbon black}} = 0.05 \text{ mg} - m_{\text{RuO}_2}$. ^dmass of carbon black added in the catalyst ink. ^eTotal mass = $m_{\text{RuO}_2} + m_{\text{Carbon black}} + m_{\text{Carbon black-ink}}$.

In response, we have revised **Supplementary Tables 3, 4**, and the related labels and descriptions in **Supplementary Figs. 24, 25 and 26** in the revised supporting information.

Original comment 4-11:

On fig 3a, it seems that the peak for the C1s is shifted to lower energy while this of Ru 3d (3d_{5/2}) does not move. My question is: how were those spectra calibrated in energy since usually the C1s peak, related to adventitious carbon most of the time, serves as the reference?

Response:

It is a visual illusion by 3D image display in original Fig. 3a. As shown in Fig. R16, C1s peaks do not shift. Accordingly, we have replaced **Fig. 3a** with Fig. R16 in the revised manuscript.

Figure R16. *In situ* Ru 3d XPS spectra recorded at applied potential during 1.00-2.00 V_{RHE}.

Original comment 4-12:

No deconvolution of the XPS spectra are presented for the Ru³⁺ and Ru⁴⁺ ratio determination and even for all the element while a lot of literature exist on that subject. This is a missing aspect. A very good reference for Ru is 10.1002/sia.5852.

Response:

Thanks for your kind suggestion. We have done deconvolution of the Ru XPS spectra (Fig. R17). In response, we have added Fig. R17 as **Supplementary Fig. 14** in the revised supporting information. Moreover, the reference the Reviewer mentioned has been cited in **the caption of Supplementary Fig. 14**.

Figure R17. *In situ* Ru 3d XPS spectra recorded at applied potential during 1.00-2.00 V_{RHE}.

Original comment 4-13:

Neutral conditions were achieved by the use of PBS buffer, how the presence of phosphate species at the surface of the catalyst (in particular on RuO₂ fig 4f) reported through in situ surface enhanced IR spectra interfere with the catalysis process?

Response:

Thanks for your comment. The possible phosphate species on catalyst surface are HPO₄²⁻ and PO₄³⁻ in neutral solution (*J. Am. Chem. Soc.* 2013, 135, 9991). According to the literature (*J. Am. Chem. Soc.* 2019, 141, 15891; *J. Phys. Chem. A* 2004, 108, 29), the P-O stretching vibration bands of HPO₄²⁻ are located at 1077 and 990 cm⁻¹, while the P-O bands of PO₄³⁻ are centered at 1006 and 850 cm⁻¹. We note that the location of these peaks is different from that of *OOH detected (Fig. 4e, f) and does not interfere with the catalysis process. Moreover, we have confirmed the *OOH bands by isotopic measurements with H₂¹⁸O according to your suggestion (Please see details in our response to your comment 4-18).

Original comment 4-14:

The description of all the materials does not seem very clear and hence rather incomplete. In particular, with the data provided, it seems that the modelling of the RuO₂/CoO_x hybrid is not in agreement with the experiments. On fig. S5, we can see a classical TEM image of Ru/CoO and EDS elemental mapping showing that Ru is dispersed throughout all the surface of the rod and so I cannot relate to the reported description of the system as islands of RuO₂ on a CoO_x surface. In particular the HAADF-STEM (and not HADDF-STEM) presented in fig. S14a shows not a significant contrast that should arise from the difference in atomic number between Ru and Co. It is true they are both converted in oxides but still. So I think a HAADF-STEM images of the Ru/CoO_x precursor is needed because I have the feeling that all the surface is covered by Ru particles and so the effect of the interface/triple point on the catalytic activity should be limited (cf paragraph “Origin of the enhanced OER activity”). The analysis of the fig 5a is deceiving in my point of view because the figures 2c and 2d are much in favor of a continuous Ru shell at the surface of the CoO rod. Could the authors provide more details on the actual description of their samples and hybrids, with more experimental evidences, since this a new material described?

Response:

Thanks for your comment. The dispersion of Ru element throughout the surface mentioned by the Reviewer results from the limited spatial resolution of EDS at low magnification (with large electron beam size). Moreover, we agree with the Reviewer that the contrast of RuO₂ and CoO_x is indeed not significant in original Supplementary Fig. 14a. This is due to the relatively low resolution of HAADF-STEM image taken during EDS mapping. Fig. R18 shows the same RuO₂/CoO_x imaged in the TEM mode when searching the sample (not the same area as that in original Supplementary Fig. 14a). As seen, RuO₂ “islands” are very clear on CoO_x.

Figure R18. TEM image of the RuO₂/CoO_x.

We note that the distribution of nanoparticles on the nanorod is three-dimensional, however the TEM/HAADF-STEM image is a two-dimensional projection. The “islands” or “continuous shell” like morphology of nanoparticles depends on the zone axis of CoO when the TEM image is taken. To explain this phenomenon, we construct a structure of Ru/CoO (Fig. R19a and b are the same model, just viewed from different directions). If the nanorod is situated in the direction shown in Fig. 19a, the “islands” can be clearly seen (This is the case of Fig. R18). If we tilt the zone axis of CoO nanorod to the direction shown in Fig. R19b to see the Ru/CoO_x interface, the projected nanoparticles look like a shell.

Figure R19. Structure model of Ru/CoO.

Moreover, we have performed additional experiments to verify that the sites at the interface are indeed the active site for OER. Please *see* details in Response to your comment 4-15. In response, we have added Fig. R18 as **Supplementary Fig. 17a** in the revised supporting information.

Original comment 4-15:

Considering the fact that interfacing of RuO₂ with CoO_x leads to Ru atoms with a higher electron density (cf fig 1d and 2e), this seems to be good for improving the stability but how is it better for improving the oxidation of water of hydroxide ions? I might miss something but Co charge does not change and the mechanism is said to occur at the interface (Ru/Co dual-atom site) between the two materials.

Response:

Thanks for your comment. In RuO₂/CoO_x, the electron interaction among the face-to-face Ru–O–Co interfacial atoms enhance the stability, while the Ru/Co dual-atom site exposed around the interface is responsible for the improved activity. Our DFT calculation reveals that the oxygen

intermediates are co-adsorbed at the Ru/Co dual-atom site (Fig. R20a). As discussed in the main text, the co-adsorption configuration facilitates the stabilization of key intermediate $*\text{OOH}$ at the Ru/Co dual-atom site around the interface and transforms the OER mechanism with a significantly decreased energy barrier compared with RuO_2 (Fig. 20b). This is verified by our electrochemical tests and KIE results, as discussed in the main text.

Figure R20. Theoretical investigation of $*\text{OOH}$ formation on (a) $\text{RuO}_2/\text{CoO}_x$ and (b) RuO_2 .

Moreover, to experimentally verify interfacial Ru/Co dual-atom sites as the most active OER sites, we have performed additional experiments. First, we deposited RuO_2 nanoparticles on carbon black with the same particle size ($\sim 2 \text{ nm}$) as that on $\text{RuO}_2/\text{CoO}_x$ (Fig. R21a and R21b). The higher current density of $\text{RuO}_2/\text{CoO}_x$ compared with that of RuO_2 with the same RuO_2 -mass (Fig. R21c) indicates the crucial role of the interface in enhancing OER performance. Moreover, we prepared $\text{RuO}_2/\text{CoO}_x$ with varied RuO_2 particle size or loading masses (Fig. R22a). Assuming either surface Ru site or interfacial Ru-Co dual-atom site as the most active sites, the corresponding site number was calculated and correlated with the OER current density at $1.50 \text{ V}_{\text{RHE}}$ (Fig. R22b and R22c). An adequate linear relationship was observed between the number of interfacial Ru-Co dual-atom site and the OER current density (Fig. R22c). These collective results reveals that the interfacial Ru/Co dual-atoms are the most active OER sites.

Figure R21. (a) TEM image of RuO₂ deposited on carbon black. (b) Polarization curves of the RuO₂ with different RuO₂ mass loadings on per cm² electrode in neutral electrolyte. (c) Comparison of current density (J) of RuO₂/CoO_x and RuO₂ at 1.50 V_{RHE} with identical RuO₂-masses.

Figure R22. (a) Polarization curves of the RuO₂/CoO_x with varied RuO₂ particle size or loading masses in neutral electrolyte. (b) and (c) Plot of current density (J) for RuO₂/CoO_x at 1.50 V_{RHE} as a function of the estimated number of surface Ru sites and interfacial Ru/Co dual-atom sites, respectively.

In response, we have added Figs. R21 and R22 as **Supplementary Figs. 34 and 36** in the revised supporting information.

Original comment 4-16:

Is there any strains or defects induced by the transformation of Ru nanoparticles into RuO₂ nanoparticles at the interface between Ru and CoO_x because we only see one detail of this interface (same fig S8, S9 and S12)?

Response:

Thanks for your comment. We constructed the interface using the lattice parameters of RuO₂ and CoO from the Crystallographic database (Fig. R23). As shown, there is good lattice match between RuO₂ and CoO. During the synthetic process, the interfacial strain was fully relaxed by treating the catalyst in N₂ flow at 400 °C for 0.5 h. We observe no evident strain and defects at the interface.

Figure R23. Interface model of RuO₂/CoO_x.

Original comment 4-17:

RuO₂ is mainly a crucial catalyst for OER in acidic conditions since very efficient PGM-free catalysts are already available for alkaline OER. I regret that there is no mention of what happens in acidic condition (and the Pourbaix Diagrams in Fig. 1a and 1b are a bit deceiving in this sense because one goes to pH = 0 and the second only to 7: the calculations are not valid when pH < 7?) Is this strategy not applicable at low pH? The system seems anyway to be a bit complex to be included in a real electrolyzing cell and dissolution of the CoO_x in low pH for instance may degrade the membrane.

Response:

Thanks for your valuable comment. We agree with the Reviewer that CoO_x can be dissolved in low pH and current RuO₂/CoO_x system cannot be directly applied in low pH environment. However, the strategy proposed in this work is applicable. Actually, we are exploring oxide support material (such as WO_x), which can be preferentially oxidized instead of RuO₂ in acidic solution. We hope to share our results in future work. Besides, we note that OER is also a very important half reaction for carbon dioxide reduction, which is preferable for operating in neutral environment (*Joule* 2021, 5, 737). However, the progress of neutral OER was far behind compared to the well-developed acidic and alkaline OER. We hope that our current developed RuO₂/CoO_x catalysts, which shows an outstanding activity and stability under neutral conditions can be applied in these occasions.

In response, we have added some outlook **in lines 18-22, page 9** in the revised conclusion. *“We believe that under the guideline built by the RuO₂/CoO_x interface, the activity and stability issues of RuO₂ in acidic environments can also be fundamentally solved by selecting appropriate support materials. We expect that this work will also contribute to future research on other renewable energy*

technology coupled with OER in neutral environments, such as reduction of carbon dioxide to multi-carbon fuels.”

Original comment 4-18:

Page 7, line 167-169, one can read “This finding is further supported by the in situ IR spectroscopy characterization (Fig. 4e and Supplementary Table 6), which shows a more pronounced HOO band of RuO₂/CoO_x in comparison with that of RuO₂ (Fig. 4f). These results suggest that the RuO₂/CoO_x exhibits a different OER mechanism compared with the pristine RuO₂.” By checking in references 33, 38 and 39 (in the SI), it seems there is no value in the literature corresponding for this vibration in the 1150-1180 cm⁻¹ range which is indicated in the supporting table 6 and highlighted in the figures 4e and 4f. So the assignment of the OOH* band seems doubtful. First the fact that the OOH* intermediate is adsorbed on RuO₂/CoO_x and not on Pt or Pd should provide a different wavelength number (it seems it is close from 1212 cm⁻¹ in reference 34 in the SI) so why the 1158 and 1177 cm⁻¹ are assigned to this specific vibration because I see no such values in the reference papers cited? Isotopic measurements with H₂¹⁸O could have been fruitful to gain evidence for this attribution. Does the phrase “the band is more pronounced” suggest that there is a quantitative difference (more or less adsorbates) and so a difference in the OER mechanism?*

Response:

Thanks for your valuable comment. Accordingly, we have listed the positions of *OOH bands in the literature in (Table R3). The difference between the wavelength of the *OOH bands in our samples and the reported values is due to the pH effect. We measured the *in situ* surface-enhanced IR spectra of CoO_x in both 1.0 M PBS and 1.0 M KOH (Fig. R24). As seen, the position of *OOH bands in 1.0 M KOH (1050 cm⁻¹) is very close to the reported value in CoO_x (Table R3). Moreover, the *OOH bands of CoO_x shift toward lower wavelength direction in alkaline solution compared with that in neutral solution (the more evident *OOH bands in alkaline solution is due to the higher OER activity, yielding more *OOH). We note that the location of *OOH on our samples in neutral solution is between the reported values of *OOH in alkaline and acidic solutions (Table R3). This is ascribed to the fact that the H atom in *OOH can form hydrogen bond with O atom in OH⁻ from electrolyte, resulting in *OOH to move in the lower wavenumber direction in IR spectra (*J. Phys. Chem. A* 2018, 122, 4481).

Table R3. Location of *OOH reported in literature.

Catalyst	Location of *OOH (cm ⁻¹)	Electrolyte	Reference
Pt/C	1212	0.1 M HClO ₄	Angew. Chem. Int. Ed. 2018, 57, 12855
Ru ₁ -Pt ₃ Cu	1212	0.1 M HClO ₄	Nat. Catal. 2019, 2, 304
CoO _x	1054	1.0 M KOH	ACS Catal. 2022, 12, 5345
NiFe MOF	1050	0.1 M KOH	Nat. Energy 2019, 4, 115

**Figure R24.** *In situ* surface-enhanced IR spectra of CoO_x in (a) 1.0 M PBS and (b) 1.0 M KOH.

According to the suggestion of the Reviewer, we have tested the *in situ* surface-enhanced IR spectra of RuO₂/CoO_x in H₂¹⁸O. As shown in Fig. R25, the *OOH band shifts to 1109 cm⁻¹ in H₂¹⁸O, which is very close to the expected value for a pure O-O stretching mode ($1158 \times \sqrt{16/18} = 1091$ cm⁻¹). This result confirms the assignment of *OOH in Fig. 4e and 4f.

Figure R25. *In situ* surface-enhanced IR spectra of RuO₂/CoO_x in neutral electrolyte prepared by (a) H₂¹⁶O and (b) H₂¹⁸O.

Yes, the more pronounced *OOH indicates the easier formation of *OOH on RuO₂/CoO_x compared with that on pristine RuO₂. This is supported by our calculation result that the rate-determining step on RuO₂/CoO_x shifts to the subsequent process (O₂ desorption).

In response, we have added Fig. R25 as **Supplementary Fig. 29** in the revised supporting information. Moreover, we have added Table R3 to replace original **Supplementary Table 6** and added a note in **Supplementary Table 6** to state that the location of *OOH bands is pH dependent. Moreover, we have added **Supplementary Note 4** to discuss the OER mechanism on RuO₂/CoO_x.

END OF RESPONSE

REVIEWER COMMENTS

Reviewer #1 (Remarks to the Author):

The authors have responded satisfactorily to the comments. The article presents a very interesting material and the work is well described. I recommend publication as is.

Reviewer #4 (Remarks to the Author):

The authors are acknowledged for the nice job they have done by addressing my remarks and by having provided good supported data. I think that this manuscript is acceptable for publication in its current state. My only regret is that for alkaline medium, the RuO₂/CoOx system was not so relevant since there already exists PGM-free catalysts for alkaline OER with similar activity. However, I expect the strategy they propose to be a nice incentive to improve the stability of OER electrocatalysts and I am more than curious to see the results of the extension of this strategy to acidic medium.

Reviewer #5 (Remarks to the Author):

(1) Overall impression:

The paper reports on an unusual (and thus interesting case) in simultaneously increasing catalytic activity and stability while reducing the amount of noble metal platinum for the oxygen evolution reaction both in neutral and alkaline conditions. The Ru is coated over a Co substrate and subsequently electrochemically oxidised to form active sites at interfacial Ru-Co “dual” atoms. Much effort has been made to use simulations and experiments to support conclusions however not all observed phenomena could be explained as this would make the article too long. The authors suggest that the combined enhancement of activity is obtained since the rate determining step during OER is different from and energetically less intense than that for either RuO or CoOx while stability is enhanced by a combination of the electronic interaction between RuO and CoOx as well as the sacrificial “oxidation” of CoOx. It is hoped that the authors are planning further studies that investigate issues like stability under dynamic current/voltage loading as well as with higher current density than 150 mA/cm² in neutral pH as well as in acidic pH. After minor revisions, it is recommended that the paper is published as it has the potential to stimulate more work in this area in the community especially regards understanding better the observed phenomena.

(2) Abstract

The concept is only vaguely described as "artificially constructed a Ru=2/CoOx interface- it is not clear what this means. Also the pH range which in the stabilisation and enhanced activity were verified is not specified. moreover, no quantitative data was provided for these verifications.

(3) Experimental approach and concept

Nowadays, sustainability and other issues are becoming important for the choice of raw materials. The authors should somewhere in the introduction comment on the replacement of Ru with Co which is also on the critical list of materials (<https://www.usgs.gov/news/national-news-release/us-geological-survey-releases-2022-list-critical-minerals>; <https://eur-lex.europa.eu/legal-content/EN/TXT/PDF/?uri=CELEX:52020DC0474&from=EN>) in view of the desired increased deployment of electrolysis in the future.

(4) Methods

The article cited by the authors as a reference for the procedure for production of the CoOx samples, itself also cites another paper (<https://doi.org/10.1038/ncomms12876>). Please update appropriately.

Why didn't the authors go higher than 150 mA/cm² in neutral solution?

What was the Ru loading of the commercial catalyst used for comparison in Supp. Fig 26?

(5) Presentation and discussion of results

The authors should clarify some assertions made in the main text based on supplementary Figs 34 & 36. Considering the wider picture, simply comparing the Ru loading with and without Co support e.g. in Supp. Fig 34, is rather simplistic in reality, one should also consider the total "active or catalytically relevant" element loading since Co seems to also play a role in the catalytic activity. Extend the caption of Supp Fig 36(a) to help the reader distinguish which particle size each of the various curves with a Ru loading of 10µg relates to.

In Supp Tables 5 & 7, it would also be useful to present stability data if possible because that is the main theme of this manuscript.

Response to Reviewer #5

Original comment 1-1:

Overall impression: The paper reports on an unusual (and thus interesting case) in simultaneously increasing catalytic activity and stability while reducing the amount of noble metal platinum for the oxygen evolution reaction both in neutral and alkaline conditions. The Ru is coated over a Co substrate and subsequently electrochemically oxidised to form active sites at interfacial Ru-Co “dual” atoms. Much effort has been made to use simulations and experiments to support conclusions however not all observed phenomena could be explained as this would make the article too long. The authors suggest that the combined enhancement of activity is obtained since the rate determining step during OER is different from and energetically less intense than that for either RuO₂ or CoO_x while stability is enhanced by a combination of the electronic interaction between RuO₂ and CoO_x as well as the sacrificial “oxidation” of CoO_x. After minor revisions, it is recommended that the paper is published as it has the potential to stimulate more work in this area in the community especially regards understanding better the observed phenomena.

Response:

We would like to thank the Reviewer for his/her positive recommendation and valuable comments to help us to improve the quality of this manuscript.

Original comment 1-2:

It is hoped that the authors are planning further studies that investigate issues like stability under dynamic current/voltage loading as well as with higher current density than 150 mA/cm² in neutral pH as well as in acidic pH.

Response:

Thanks for your kind suggestion. Accordingly, we have tested the stability of RuO₂/CoO_x with varied current densities from 10 to 100 mA cm⁻² in 1.0 M PBS (Fig. R1). As shown, RuO₂/CoO_x shows excellent dynamic stability. Moreover, following the suggestion of the Reviewer, we have increase the loading mass of RuO₂/CoO_x to achieve higher current density. Please *see* our response to your comment 4-2.

Figure R1. Dynamic stability test of RuO₂/CoO_x with current densities from 10 to 100 mA cm⁻² in 1.0 M PBS.

In response, we have added Fig. R1 as **Supplementary Fig. 19** in the revised supporting information. Moreover, the above related discussion on the dynamic stability of RuO₂/CoO_x has been added **in lines 22-24, page 6** in the revised manuscript.

“Significantly, the RuO₂/CoO_x catalyst works stably at a constant current density of 10 mA cm⁻² for more than 200 h (Fig. 3f), and affords an excellent dynamic stability with varied current density from 10 to 100 mA cm⁻² (Supplementary Fig. 19).”

Original comment 2:

Abstract: The concept is only vaguely described as “artificially constructed a RuO₂/CoO_x interface- it is not clear what this means. Also the pH range which in the stabilisation and enhanced activity were verified is not specified. Moreover, no quantitative data was provided for these verifications.

Response:

Thanks for your valuable comment. Accordingly, Abstract in the revised manuscript has been rewritten **in lines 20-22, page 1,**

“Here, we report breaking the stability and activity limits of RuO₂ in neutral (pH=7) and alkaline (pH=14) environments by constructing a RuO₂/CoO_x interface.”

and **in lines 1-3, page 2.**

“As a result, the RuO₂/CoO_x catalyst achieves a record high OER activity, affording an ultra-low overpotential of 0.24 and 0.16 V to drive an OER current density of 10 mA cm⁻² in both neutral and alkaline solutions, respectively.”

Original comment 3:

Experimental approach and concept: Nowadays, sustainability and other issues are becoming important for the choice of raw materials. The authors should somewhere in the introduction comment on the replacement of Ru with Co which is also on the critical list of materials (<https://www.usgs.gov/news/national-news-release/us-geological-survey-releases-2022-list-critical-minerals>; <https://eur-lex.europa.eu/legal-content/EN/TXT/PDF/?uri=CELEX:52020DC0474&from=EN>) in view of the desired increased deployment of electrolysis in the future.

Response:

Following the suggestion of the Reviewer, one new sentence has been added **in lines 2-3, page 3** in the revised Introduction to discuss reducing the use of Ru for the wide application of electrolysis in the future.

“Moreover, the interface construction may use some cost-effective materials to reduce the use of precious metal Ru and achieve sustainable water electrolysis.”

Original comment 4-1:

Methods: The article cited by the authors as a reference for the procedure for production of the CoO_x samples, itself also cites another paper (<https://doi.org/10.1038/ncomms12876>). Please update appropriately.

Response:

Thanks for your suggestion. The reference the Reviewer mentioned has been added as **Reference [38]** in the Methods and in the Reference list in the revised manuscript.

Original comment 4-2:

Why didn't the authors go higher than 150 mA/cm² in neutral solution?

Response:

Thanks for your valuable suggestion. We have used nickel foam to load RuO₂/CoO_x with a higher mass of 1.5 mg cm⁻². As shown in Fig. R2, the RuO₂/CoO_x can achieve a current density up to 400

mA cm^{-2} at $1.92 V_{\text{RHE}}$ in neutral solution.

Figure R2. OER polarization curve of $\text{RuO}_2/\text{CoO}_x$ with a catalyst loading of 1.5 mg cm^{-2} in neutral electrolyte (1.0 M PBS).

In response, we have added Fig. R2 as **Supplementary Fig. 29** in the revised supporting information. Moreover, the above related discussion has been added **in lines 10-11, page 7** in the revised manuscript.

“Besides, the current density of $\text{RuO}_2/\text{CoO}_x$ can achieve 400 mA cm^{-2} at $1.92 V_{\text{RHE}}$ when the mass of $\text{RuO}_2/\text{CoO}_x$ catalyst on nickel foam is increased to 1.5 mg cm^{-2} (Supplementary Fig. 29).”

Original comment 4-3:

What was the Ru loading of the commercial catalyst used for comparison in Supp. Fig 26?

Response:

Thanks for your kind noticing. The loading mass of RuO_2 for comparison in Supplementary Fig. 27 (original Supplementary Fig. 26) is 0.255 mg cm^{-2} . In response, we have specified the loading mass of RuO_2 in the **caption of Supplementary Fig. 27**.

“It shows that the catalyst with RuO_2 loading of $84 \mu\text{g cm}^{-2}$ exhibits the optimal OER activity, which is better than the commercial RuO_2 catalyst (0.255 mg cm^{-2}).”

Original comment 5-1:

Presentation and discussion of results: The authors should clarify some assertions made in the main

text based on supplementary Figs 34 & 36. Considering the wider picture, simply comparing the Ru loading with and without Co support e.g. in Supp. Fig 34, is rather simplistic in reality, one should also consider the total “active or catalytically relevant” element loading since Co seems to also play a role in the catalytic activity.

Response:

Thanks for your kind suggestion. Accordingly, we have compared the current densities of CoO_x, and RuO₂/CoO_x, RuO₂ with varied RuO₂-masses at 1.65 V_{RHE} (Fig. R3a). Moreover, we have added the polarization curve of the CoO_x in Fig. R3b. These results show more clearly the role of pristine CoO_x, RuO₂ and RuO₂/CoO_x interface in catalysing OER, as suggested by the Reviewer.

Figure R3. (a) Comparison of current density (J) of CoO_x, RuO₂/CoO_x and RuO₂ at 1.65 V_{RHE}. Note that the data for RuO₂/CoO_x and RuO₂ with different RuO₂-masses are presented. (b) Polarization curves of the RuO₂/CoO_x with varied RuO₂ particle size or loading masses and CoO_x in neutral electrolyte. We note that the orange, blue, and dark brown curves represent RuO₂/CoO_x (10 μg_{RuO₂}) with RuO₂ particle size of 2, 3 and 4 nm, respectively.

In response, Fig. R3a and Fig. R3b was added as **Supplementary Fig. 37b** and **Supplementary Fig. 39a** in the revised supporting information. Moreover, the related discussion have been added in the **caption of Supplementary Fig. 37**.

“Note that the current densities of RuO₂/CoO_x are significantly higher than those of RuO₂ and CoO_x, indicating that the RuO₂/CoO_x interface is the key to improve the OER performance of RuO₂/CoO_x.”

Original comment 5-2:

Extend the caption of Supp Fig 36(a) to help the reader distinguish which particle size each of the various curves with a Ru loading of 10μg relates to.

Response:

Thanks for your kind suggestion. In response, we have added the relative description suggested by the Reviewer in the **caption of Supplementary Fig. 39a** (original Supplementary Fig. 36a) in the revised supporting information.

“We note that the orange, blue, and dark brown curves represent RuO₂/CoO_x (10 μgRuO₂) with RuO₂ particle size of 2, 3 and 4 nm, respectively.”

Original comment 5-3:

In Supp Tables 5 & 7, it would also be useful to present stability data if possible because that is the main theme of this manuscript.

Response:

As suggested, stability data have been added in Tables R1 and R2. In response, **Supplementary Tables 5 and 7** have been updated with Tables R1 and R2 in the revised supporting information.

Table R1. Performance comparison between RuO₂/CoO_x catalyst and the reported highly active catalysts in neutral solutions.

Catalyst	Electrolyte	Overpotential @ 10 mA cm ⁻² (mV)	Loading (mg cm ⁻²)	TOF (s ⁻¹)	Current density retention after stability test	Reference
RuO₂/CoO_x	1.0 M PBS	242	0.25	1.21@300 mV 3.61@400 mV	99.8% after 200 h@10 mA cm⁻²	This work
RuIrCaO _x	0.50 M KHCO ₃	250	0.42	0.36@400 mV	99.3% 200 h@10 mA cm ⁻²	16
RuO ₂ @C	1.0 M PBS	269	0.76	--	--	17
PdP ₂ /C	1.0 M PBS	277	0.28	--	--	18
Ir-NSG ^a	1.0 M PBS	307	0.30	--	97.3% after 4.2 h@1.4 V	19
IrRu@Te ^b	1.0 M PBS	309	0.60	--	--	20
RhCo	1.0 M PBS	310	2.00	--	94.6% after 5 h@1.6 V	21
N-Fe ₂ PO _{5-x} ^c	1.0 M PBS	315	2.50	--	99.6% after 30 h@50 mA cm ⁻²	22
NiCoFeP	0.50 M KHCO ₃	330	0.39	--	97.4% after 100 h@10 mA cm ⁻²	23
Ru-RuO ₂ /C ₃ N ₄	1.0 M PBS	342	0.14	0.03@300 mV	97.5% after 5.5 h @10 mA cm ⁻²	24

Ir ₁ -Co(OH) ₂ ^d	1.0 M PBS	373	0.57	--	98.5% after 10 h@10 mA cm ⁻²	25
Co ₃ O ₄	3.0 M KPi	390	0.16	0.01@400 mV	--	26
Co-Pi/Ti ^c	1.0 M PBS	450	0.95	0.07@410 mV	99.4% after 20 h@10 mA cm ⁻²	27
Co-MOF ^f	0.10 M PBS	548@2 mA cm ⁻²	0.71	0.03@400 mV	--	28
NiFe ₂ O ₄ /FeNi ₂ S ₄	0.20 M PBS	253@1 mA cm ⁻²	0.20	--	--	29
Co ₄ Mo	0.10 M PBS	490	0.20	0.03@490mV	97.9% after 15 h@10 mA cm ⁻²	30

Table R2. Performance comparison between RuO₂/CoO_x catalyst and the reported highly active catalysts in alkaline solutions.

Catalyst	Electrolyte	Overpotential @ 10 mA cm ⁻² (mV)	Loading (mg cm ⁻²)	TOF (s ⁻¹)	Current density retention after stability test	Reference
RuO₂/CoO_x	1.0 M KOH	165	0.25	8.75@300 mV	97.3% after 20 h@1 A cm ⁻²	This work
NiFe-Boride	1.0 M KOH	167	0.39	--	99.7% after 100 h@20 mA cm ⁻²	36
(Ru-Co)O _x	1.0 M KOH	171	0.80	0.25@270 mV	98.1% after 10 h@10 mA cm ⁻²	37
Ir/CoNiB	1.0 M KOH	178	--	0.37@300 mV	98.4% after 50 h @ 100 mA cm ⁻²	38

NiO/NiFe LDH ^a	1.0 M KOH	180	0.20	0.71@300 mV	99.8% after 10 h @20 mA cm ⁻²	39
NiFeCu	1.0 M KOH	180	0.45	--	99.7% after 20 h @20 mA cm ⁻²	40
NiMO _x /NiMoS	1.0 M KOH	186	--	--	98.6% after 25 h@500 mA cm ⁻²	41
LNF ^b	1.0 M KOH	189	0.23	--	98.7% after 100 h@10 mA cm ⁻²	42
FeCoW	1.0 M KOH	223	0.21	0.46@300 mV	99.8% after 550 h@30 mA cm ⁻²	43
Ru/CoFe-LDH	1.0 M KOH	198	1.0	--	99.5% after 24 h@200 mA cm ⁻²	14
Mo-Co ₉ S ₈	1.0 M KOH	200	1.0	--	99.7% after 72 h@10 mA cm ⁻²	44
Ru ₁ -FeCoNi	1.0 M KOH	205	0.25	--	98% after 48 h@10 mA cm ⁻²	45
CoV-Fe _{0.28} ^c	1.0 M KOH	215	0.28	--	96.9% after 40 h@1.55 V	46
Co ₃ O ₄ /Fe _{0.33} Co _{0.66} P	1.0 M KOH	215@50 mA cm ⁻²	2.5	--	89.4% after 150 h@240 mV	47
NiFeP	1.0 M KOH	218	--	--	98.1% after 30 h@10 mA cm ⁻²	48
Fe-CoP/CoO	1.0 M KOH	219	0.29	--	97.5% after 12 h@10 mA cm ⁻²	49

NiFe LDH ^d	1.0 M KOH	225	0.10	--	--	50
NiFeRu-LDH	1.0 M KOH	225	1.20	--	98.8% after 10 h@10 mA cm ⁻²	51
CoFeWO _x	1.0 M KOH	231	0.20	0.54@300 mV	99.4% after 120 h@100 mA cm ⁻²	52
Au ₁ -NiFe LDH	1.0 M KOH	237	2.0	--	91.2% after 20 h@100 mA cm ⁻²	53
NiFe-LDH	1.0 M NaOH	240	--	--	--	54
Fe _{0.4} Co _{0.6} Se ₂	1.0 M KOH	270	0.50	1.23@300 mV	98.3% after 24 h@10 mA cm ⁻²	55
Ir ₁₆ -PdCu	0.1 M KOH	284	0.050	64.1@300 mV	98.6% after 10 h@10 mA cm ⁻²	56
RuO ₂	0.5 M KOH	358	0.025	0.53@400 mV	--	57

END OF RESPONSE